# Oxidation of Archean upper mantle caused by crustal recycling

Lei Gao[1,2], Shuwen Liu [2✉], Peter A. Cawood [3✉], Fangyang Hu[4,5], Jintuan Wang[6], Guozheng Sun[2,7] & Yalu Hu[2]

The redox evolution of Archean upper mantle impacted mantle melting and the nature of chemical equilibrium between mantle, ocean and atmosphere of the early Earth. Yet, the origin of these variations in redox remain controversial. Here we show that a global compilation of ~3.8-2.5 Ga basalts can be subdivided into group B-1, showing modern mid-ocean ridge basalt-like features ($(Nb/La)_{PM} \geq 0.75$), and B-2, which are similar to contemporary island arc-related basalts ($(Nb/La)_{PM} < 0.75$). Our V-Ti redox proxy indicates a more reducing upper mantle, and the results of both ambient and modified mantle obtained from B-1 and B-2 samples, respectively, exhibit a ~1.0 log unit increase in their temporal evolution for most cratons. Increases in mantle oxygen fugacity are coincident with the changes in basalt Th/Nb ratios and Nd isotope ratios, indicating that crustal recycling played a crucial role, and this likely occurred either via plate subduction or lithospheric drips.

[1] Key Laboratory of Geological Processes and Mineral Resources, School of Earth Sciences and Resources, China University of Geosciences, Beijing 100083, PR China. [2] Key Laboratory of Orogenic Belts and Crustal Evolution, Ministry of Education, School of Earth and Space Sciences, Peking University, Beijing 100871, PR China. [3] School of Earth, Atmosphere and Environment, Monash University, Melbourne, VIC 3800, Australia. [4] Key Laboratory of Mineral Resources, Institute of Geology and Geophysics, Chinese Academy of Sciences, Beijing 100029, PR China. [5] Innovation Academy for Earth Science, Chinese Academy of Sciences, Beijing 100029, PR China. [6] State Key Laboratory of Isotope Geochemistry, Guangzhou Institute of Geochemistry, CAS, Guangzhou 510640, PR China. [7] Key Lab of Submarine Geosciences and Prospecting Techniques, MOE and College of Marine Geosciences, Ocean University of China, Qingdao 266100, PR China. ✉email: swliu@pku.edu.cn; peter.cawood@monash.edu

Redox state of the Archean upper mantle buffered atmospheric composition and influenced the early ocean-atmosphere system through the geochemical behavior of volatile elements (e.g., S, C and N)[1–3]. Previous investigations of the mantle redox state have focused on the $Fe^{3+}/\Sigma Fe$ content in pristine mantle and mantle-derived rocks[1,4,5], the behavior of redox-sensitive elements (e.g., V) in komatiites[6], and the V-Sc (Sc is a typical redox-insensitive element) redox proxy in Archean basalts[7,8]. These methods provide an in principle estimate of oxygen fugacity ($fO_2$; relative to the fayalite-magnetite-quartz buffer ($\Delta FMQ$)) of the upper mantle. However, in terms of Archean samples, the $Fe^{3+}/\Sigma Fe$ content of minerals and glasses might be readily modified during alteration, metamorphism and syn-eruptive process[6,9,10]. The behavior of V in komatiites shows a significant increase of ~1.3 log units in mantle $fO_2$ between ~3.5 Ga and 1.9 Ga[6]. This change, however, might not reflect the nature of the upper mantle because komatiites constitute less than 10% of exposed Archean volcanic rocks, and their magmatic sources are thought to be complicated with materials from either the core-mantle or the upper-lower mantle boundaries[11,12]. Compared to komatiites, basalts constitute the dominant component of Archean continental crust (~65–75 %) and can provide alternative insights into resolving $fO_2$ of the mantle[1,8,13]. Some studies indicated that the mantle $fO_2$ has been at present-day levels since ~3.9 Ga[7], but others inferred a dominantly reducing upper mantle on the basis of the significantly lower V/Sc ratios in Archean mid-ocean ridge basalt (MORB)-like samples relative to their modern counterparts[1,8]. The reason of these contrasting interpretations is attributed to choices of different mantle melting reactions and partition coefficients (Kds) in converting V/Sc ratios to mantle $fO_2$[1,7,8]. Our recent study found that the Kds of V, Sc and Ti are influenced not only by melting temperature-pressure (P-T) and $fO_2$, but also by the compositions of mantle mineral and water contents[14], which have not been fully considered in previous $fO_2$ estimations. Thus, suitable Kds and mantle melting reactions are critical for the accurate estimation of $fO_2$. In addition, although Archean MORB-like basalts or eclogites are increasingly used in mantle $fO_2$ calculations, limited investigations of basalts with characteristics indicative of lithospheric recycling (e.g., modern island arc basalt (IAB)) have impeded a comprehensive understanding of the early Earth's mantle-crust interactions[15–18].

We have assembled a global whole-rock geochemical database of Archean (~3.8–2.5 Ga) basalts ($N = 2304$; See more details of data collection and filtration in Methods) from fourteen cratons (Supplementary Fig. 1). They exhibit a significant compositional diversity, and are divided into Basalt-1 (B-1) and Basalt-2 (B-2) groups with modern MORB- and IAB-like chemical features, respectively. We applied an updated V-Ti redox proxy, which may be more sensitive than V-Sc systematics because of the more incompatible character of Ti during mantle melting and the influence of Sc content by garnet residues[14], to this database. The results from applying this proxy are: (1) The average V/Ti ratios of Archean basalts are relatively lower than those of modern counterparts, and when converting them to mantle $fO_2$, the Archean upper mantle was more reducing than it is today; (2) There is a marked difference in redox state between ambient ($\Delta FMQ$ $-1.31 \pm 0.58$ (1 SD)) and modified mantle ($\Delta FMQ$ $-0.88 \pm 0.84$ at 2 wt.% $H_2O$ and $\Delta FMQ$ $-0.39 \pm 0.89$ at 4 wt.% $H_2O$) obtained from B-1 and B-2 magmas, respectively; (3) Both the ambient and modified mantle have commonly undergone significant oxidization with ~1.0 log unit increase of most Archean cratons and possible craton groups; and (4) Changes in mantle $fO_2$ are closely associated with changes in basalt Th/Nb ratios and Nd isotope ratios that are sensitive to crustal recycling.

## Results

**Geochemical characteristics of Archean basalt.** Based on the established tectonic settings, modern basalts are commonly divisible into non-subduction and subduction-related types. The former records the ambient mantle conditions via adiabatic decompression melting, and the latter indicates fluid flux melting due to crustal/lithospheric recycling[19]. Our statistical analysis of modern basalts suggests that $(Nb/La)_{PM} \geq 0.75$ (PM, primitive mantle normalized[20]) is a powerful criteria identifying primitive melts that were generated in the non-subduction settings (e.g., midocean ridge, oceanic island and oceanic plateau), whereas the crustal recycling-related basaltic melts generally have $(Nb/La)_{PM} < 0.75$ (Supplementary Fig. 2). In previous studies, the differentiated degree of rare earth elements (REEs; e.g., La/Sm and La/Yb ratio) was taken as the main criteria to distinguish Archean basalts with crustal recycling-related origins[21]. However, on modern Earth, some initial arc basalts and most of the back-arc basin basalts (BABBs) also show low tholeiite-like La/Yb ratios and similar REE patterns to those of modern tholeiitic MORBs (Data from the GEOROC and Schmidt's database)[19]. Therefore, on this basis, we subdivide the Archean basalts into B-1 and B-2 groups with $(Nb/La)_{PM} \geq$ and < 0.75, respectively.

The B-1 samples ($N = 797$) predominantly belong to the tholeiitic rock series according to their low La/Yb ratios of 0.50-3.73 (La/Yb $\leq$ 2.6 for tholeiitic and La/Yb $\geq$ 5.3 for calc-alkaline rock series[22]; Supplementary Fig. 3). They are characterized by low $SiO_2$ contents (average 49.94 wt.%), and high MgO (average 8.41 wt.%) and TFeO (average 12.01 wt.%) contents, corresponding to an average Mg# value of 58 (Supplementary Table 1). They have light rare earth element (LREE) depleted to nearly flat patterns with an average $(La/Yb)_N$ ratio of 1.04 (Supplementary Table 1), similar to modern MORBs.

The B-2 samples ($N = 1507$) have similar major oxide features to those of B-1 with average $SiO_2$ of 50.60 wt.%, MgO of 8.33 wt.%, TFeO of 11.23 wt.% and Mg# of 59. In terms of REE ratios, they are intermediate between tholeiitic and calc-alkaline rock series due to a wide range in La/Yb ratios (up to 72)[22], clearly distinct from B-1 samples (Supplementary Fig. 3). Some B-2 samples have LREE depleted to flat patterns, similar to the modern basalts in the initial arc and back-arc basin settings[19], but most of them display LREE enriched patterns with average $(La/Yb)_N$ of 2.19 (Supplementary Table 1).

**Redox state of Archean upper mantle.** The principle of the V-Ti redox proxy is that V is a multivalent element (2+, 3+, 4+, and 5+), and its Kds vary with $fO_2$, melting P-T conditions, water contents and mineral compositions, in contrast the Kds of Ti are independent of $fO_2$[7,8,14]. In addition, both V and Ti are generally immobile during dehydration and melting of oceanic crust (including sediments), and are not strongly enriched in continental crust[7,14]. Together with their highly incompatible features among most mantle minerals, the V-Ti systematics can 'see through' early magmatic differentiation. Based on a thorough consideration of peridotite melting reactions and water contents, the V-Ti redox proxy has successfully estimated the $fO_2$ of both modern ambient and modified mantle[14], providing a viable solution to the mantle $fO_2$ debate in the early Earth.

Recent studies highlight that degassing or interaction with polyvalent gas species (e.g., S) of basalts can potentially change the $Fe^{3+}/\Sigma Fe$ contents, leading to an incorrect estimation of redox state inferred for their magmas[9,10]. However, there is no evidence that these changes would affect the V/Ti ratio, which is an indicator to the mantle source of less- or undifferentiated magmas[1]. Therefore, in this study, we only use basalts with MgO $\geq$ 8.5 wt.% to avoid possible effects from the late overprint of

fractional crystallization and contamination from continental material[23].

Before determining the mantle $f$O$_2$, we use the Fractionated-PT software to calculate melting P-T conditions for the parental magmas of basalts[23] (Supplementary Table 1; Supplementary Fig. 4). Due to the lack of water content assessment for Archean samples, we tentatively used the modern values of MORBs (~0.2 wt.%) and IABs (~2-4 wt.%) to represent those of Archean B-1 and B-2 samples, respectively[24,25]. The results indicate that B-1 samples were derived at high temperatures (T = ~1359–1628 °C) and pressures (P = ~0.8–3.8 GPa), corresponding to a potential temperature (T$_P$) of ~1590 °C (Supplementary Fig. 4a). B-2 samples were derived at T = ~1251–1563 °C and P = ~0.8-3.5 GPa at 2 wt.% H$_2$O and T = ~1215–1519 °C and P = ~0.8–3.3 GPa at 4 wt.% H$_2$O[23] (Supplementary Fig. 4b, c). Based on these calculations, the melting P-T conditions of B-1 samples generally straddle the phase transition zone (P = ~2.2 GPa) between anhydrous spinel- and garnet-facies[23,26]. We therefore employ near fractional melting to perform both low (~1 GPa; Model A) and high pressure (~3 GPa; Model B) peridotite melting models for B-1 samples, and ~1-2 GPa (Model C) and ~3 GPa (Model D) melting models for B-2 samples under hydrous condition[27–31] (Supplementary Table 2; Methods).

Sources of uncertainties that arise from trace element analyses, mantle compositional heterogeneity, P-T determinations and systematic biases for the Kds of V in mantle minerals and the $f$O$_2$ are incorporated into the application of V-Ti redox proxy and conversion of V/Ti ratio into the mantle $f$O$_2$. These uncertainties are incorporated as the following: (1) Analytical uncertainties of V and Ti are set to 5 %, assuming that all of samples collected in this study were analyzed by the inductively-coupled plasma mass spectrometry (ICP-MS) technique[6,14]; (2) Uncertainties of mantle heterogeneity were taken as 7 % for V and 12 % for Ti[14,31]; (3) Uncertainties of P-T determinations are based on an absolute value of 0.20 GPa and 3 %, respectively[23], assuming that there were no additional uncertainties in the water content assessment; (4) Estimate of uncertainties and systematic biases for the functions between Kds of V in mantle minerals and the $f$O$_2$ were incorporated into further propagations[14]. In this study, uncertainties (1) and (2) were firstly propagated into the calculation of the compositions of V and V/Ti in the melts and their Kds in bulk rock compositions, which were then propagated to the Kds in mantle minerals and $f$O$_2$ on the basis of a full account of uncertainties (3) and (4). On this basis, the average propagated 1 SD uncertainties of the mantle $f$O$_2$ revealed by B-1 and -2 samples are 0.40 and 0.32 log units, respectively (see the black error bar in Figs. 1–3).

The B-1 samples have variable V (143–507 ppm) and Ti (2619–9000 ppm) contents with an average 100*V/Ti ratio of 5.45 ± 1.17 (1 SD), significantly higher than those of modern MORBs (3.61 ± 0.77[14]; Supplementary Table 1; Supplementary Fig. 5a). However, the B-2 samples are characterized by lower V (94–440 ppm) and Ti (2319–7740 ppm) contents, showing an average 100*V/Ti ratio of 5.49 ± 1.09 (1 SD), close to those of modern IABs (4.75 ± 1.90[14]; Supplementary Table 1; Supplementary Fig. 5b). When converting V-Ti systematics to the mantle $f$O$_2$, the results of ambient mantle indicated by B-1 magmas exhibit an average value of ΔFMQ −1.31 ± 0.58 (1 SD; Fig. 1), slightly lower than those obtained by V-Sc redox proxy (ΔFMQ −1.19 ± 0.17)[1,8]. There is also a significant $f$O$_2$ difference between B-1 and B-2 samples, indicative of the significant redox heterogeneity between ambient and modified mantle (Figs. 1 and 2). The modified mantle of B-2 samples is more oxidized than Archean ambient mantle with average ΔFMQ −0.88 ± 0.84 at 2 wt.% H$_2$O (Fig. 2) and ΔFMQ −0.39 ± 0.89 at 4 wt.% H$_2$O (Fig. 3). Comparing our results to the redox state of modern MORBs (ΔFMQ −0.03 ± 0.38) and IABs (ΔFMQ + 0.42 ± 0.63 at

2 wt.% H$_2$O and ΔFMQ + 0.82 ± 0.62 at 4 wt.% H$_2$O) obtained by the previous V-Ti redox proxy[14], we found that the Archean upper mantle was intrinsically more reducing than it is today (Figs. 1–3).

## Discussion

Based on the estimated $f$O$_2$ results, the values of modified mantle beneath the North Atlantic Craton increase by ~1.2 log units from ΔFMQ −1.58 ± 0.74 (1 SD) to ΔFMQ −0.33 ± 0.93 at 2 wt.% H$_2$O between ~3.8 Ga and 3.0 Ga (Fig. 2a). The ambient mantle beneath the North Atlantic Craton has also been oxidized from ΔFMQ −1.64 ± 0.40 to ΔFMQ −0.56 ± 0.16 (Fig. 1a). Similarly, the $f$O$_2$ values of the Pilbara Craton with ambient and modified mantle increase from ΔFMQ −1.64 ± 0.12 to ΔFMQ −0.77 ± 0.20 and ΔFMQ −1.57 ± 0.59 to ΔFMQ −0.21 ± 0.39 at 2 wt.% H$_2$O between ~3.5 and 3.2 Ga, respectively (Figs. 1 and 2). For the Kaapvaal Craton, the modified mantle values derived from ΔFMQ increase from −1.39 ± 0.57 to -0.20 ± 0.30 at 2 wt.% H$_2$O between ~3.4 and 3.0 Ga (Fig. 2). These featured $f$O$_2$ values indicate that the North Atlantic, Pilbara and Kaapvaal cratons may constitute a ~3.8–3.0 Ga craton group with a similar evolutionary history.

During the evolution of the Superior Province between ~3.0 and 2.7 Ga, the $f$O$_2$ values of its modified mantle increase from a low threshold (ΔFMQ −1.31 ± 0.66) to a high threshold (ΔFMQ −0.47 ± 0.54) at 2 wt.% H$_2$O (Fig. 2a). Similarly, the ambient mantle beneath both the Superior Province and the Yilgarn Carton have been oxidized between ~2.9 Ga and 2.7 Ga from ΔFMQ −1.79 ± 0.48 to ΔFMQ −1.16 ± 0.52 and ΔFMQ −1.76 ± 0.42 to ΔFMQ −1.29 ± 0.41, respectively (Fig. 1a). Although the amount of data is relatively limited, the $f$O$_2$ values for the Tanzania and Zimbabwe cratons fall within this ~3.0–2.7 Ga mantle oxidation trend (Figs. 1 and 2), suggesting that along with the Superior and Yilgarn cratons, these four cratons may also constitute a craton group and probably share a similar Archean evolutionary history.

Only the North China and Dharwar cratons displaying significant ~2.8–2.5 Ga magmatism[32,33]. The upper mantle beneath these cratons also display a gradual oxidation. The values of ambient mantle increased from ΔFMQ −2.14 ± 0.09 to ΔFMQ −1.09 ± 0.25 beneath the North China Craton, and from ΔFMQ −1.61 ± 0.38 to ΔFMQ −0.88 ± 0.24 beneath the Dharwar Craton (Fig. 1a), consistent with the oxidation of modified mantle from ΔFMQ −1.31 ± 0.51 to ΔFMQ −0.44 ± 0.87 and ΔFMQ −1.51 ± 0.44 to ΔFMQ −0.30 ± 0.33, respectively (Fig. 2a).

Therefore, the upper mantle (both ambient and modified mantle) beneath most Archean cratons, involving three temporal groupings has undergone significant oxidation during their evolutionary histories. Notably, the starting values of mantle oxidation beneath each craton or craton group are roughly the same (ambient mantle at approximate ΔFMQ −1.6 to −2.0 and modified mantle at ΔFMQ −1.3 to −1.1 at 2 wt.% H$_2$O; Figs. 1 and 2), indicating that the mantle redox state might have attained a relatively homogeneous level prior to 3.8 Ga. Although the final values of ambient mantle for each craton age group are variable (ΔFMQ −1.2 to −0.5; Fig. 1a), the values of modified mantle are close (ΔFMQ −0.6 to −0.4 at 2 wt.% H$_2$O; Fig. 2a), suggesting that the redox state of Archean mantle might be an effective indicator of craton maturation and cratonization.

At present, the possible three mechanisms for Archean upper mantle oxidization are: (1) Recycling of crustal material into mantle traced by N, Ca, Si, Li and B isotopic systematics[34–37]; (2) Venting of O$_2$ from the crystallization of the inner core[38,39]; and (3) Convection-derived homogenization of early high $f$O$_2$ mantle blocks generated by Fe$^{2+}$ disproportionation into Fe metal and

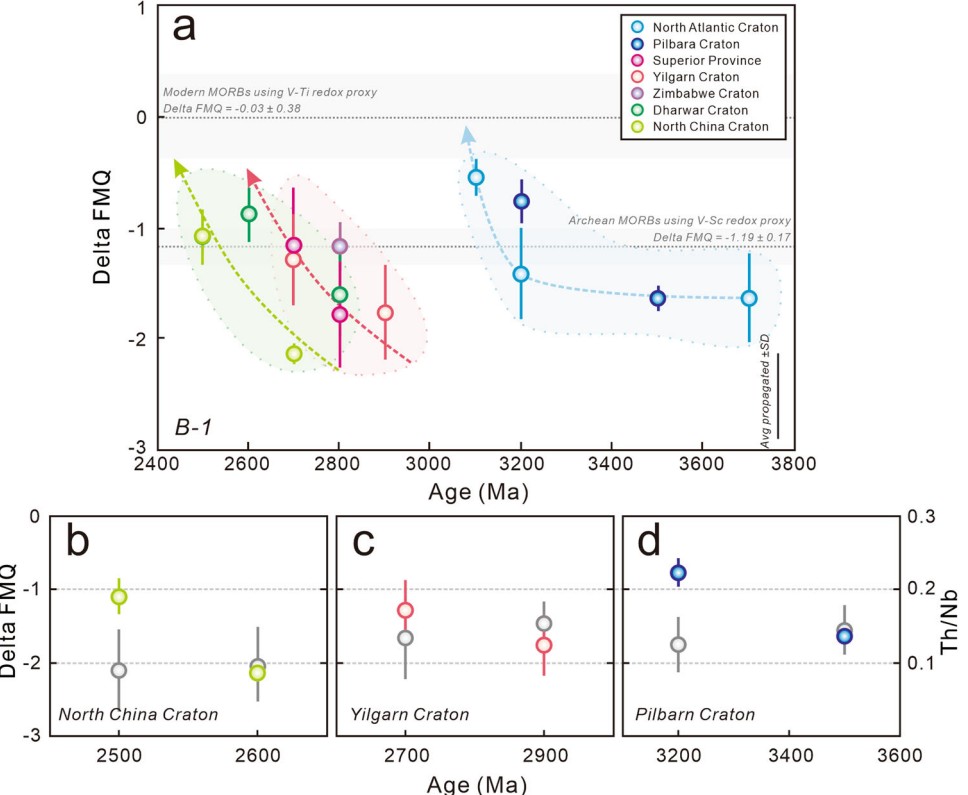

**Fig. 1 Redox state of Archean Basalt-1 (B-1) samples calibrated by the updated V-Ti redox proxy. a** The $fO_2$ estimations of B-1 samples. The $fO_2$ values and respective uncertainties (grey intervals) of Archean mid-ocean ridge basalt (MORB)-like samples estimated by V-Sc redox proxy and modern MORBs by V-Ti redox proxy were used for comparison[1,8,14]. The uncertainties in the bottom right corner represent the average 1 standard deviation (SD) propagation of B-1 samples. **b–d** The time evolution of $fO_2$ estimations and Th/Nb ratios in the North China, Yilgarn and Pilbara cratons, respectively. The grey symbols are the corresponding Th/Nb ratios of B-1 samples. Error bars show the 1 SD of the means.

$Fe^{3+}$ with a corresponding release of $O_2$ at lower mantle[2,6]. It is believed that all these mechanisms are mutually responsible for Archean mantle oxidation. However, if mechanism (2) is dominant, the starting mantle $fO_2$ values of most cratons should gradually rise over time, distinct to our discoveries that initial $fO_2$ remain unchanged (Figs. 1–3). Furthermore, Archean mantle convection is commonly considered to be sluggish[40,41], suggesting that mechanism (3) was not a dominant factor. Therefore, it is crucial to explore the relationship between crustal recycling and mantle oxidation.

Basalt Th/Nb ratios are generally taken to reflect the interaction between mantle-derived magmas and crustal materials, namely, crustal recycling, due to higher elemental abundances of Th against Nb in the continental crust[42–44]. Notably, the Th/Nb ratios of B-2 samples from each craton show a covariant elevated trend with the increase of mantle $fO_2$. For example, the North Atlantic Craton increases from 0.28 ± 0.14 (1 SD) to 0.37 ± 0.12 between ~3.8 and 3.0 Ga, the Superior Province from 0.16 ± 0.02 to 0.30 ± 0.22 between ~3.0 and 2.7 Ga and the Dharwar Craton from 0.23 ± 0.19 to 0.37 ± 0.09 between ~2.7 and 2.6 Ga (Supplementary Table 1; Fig. 2b-d). These increasing values suggest a significant increase in crustal recycling. Similarly, the εNd (t) values of B-2 samples from the Superior, Yilgarn, Tanzania and Zimbabwe cratons exhibit a significant mantle enrichment process between 3.0 Ga and 2.7 Ga[45] (Supplementary Fig. 6a; Data from the GEOROC database). Given that the Kds of Th are significantly lower than those of Nb within most mantle minerals, there should be a sharp decrease in Th/Nb ratio of mantle residues and hence B-1 samples during the melting and depletion of the ambient mantle (Supplementary Fig. 6b). However, the B-1 samples from most cratons show a nearly constant Th/Nb trend under the background of gradual oxidation of ambient mantle, for example, the ~3.5–3.2 Ga Pilbara Craton at ~0.13, the ~2.9–2.7 Ga Yilgarn Craton at ~0.14 and the ~2.7–2.5 Ga North China Craton at ~0.10 (Supplementary Table 1; Fig. 1b-d), indicating that recycled crustal materials can also influence the chemical compositions of ambient mantle perhaps via increasing mantle convection with craton maturation[46]. These processes would offset the decrease of basalt Th/Nb ratio during the depletion of the ambient mantle.

Two main mechanisms have been proposed to convey crustal materials into the hotter mantle of the early Earth: a form of plate subduction (or hot subduction)[17,47–54] and lithospheric drips[55,56]. Although our data alone cannot differentiate between these mechanisms, it does establish that Archean crustal recycling was widespread and ongoing within each craton over hundreds of millions of years [34,37].

## Methods
**Data collection and filtration.** The whole-rock geochemical database of the ~3.8-2.5 Ga basalts is a combination of the GEOROC database (http://georoc.mpch-mainz.gwdg.de/georoc/) and a subset of the North China Craton volcanic database assembled by our research group. In this study, the principles of data filtration are listed as follows: (1) We define basalts as volcanic samples with $SiO_2 = 45$–54 wt.% (including some basaltic andesites, because the primitive melts generated by melting of mantle peridotite may be more siliceous, especially in modern arc-related settings[19]) and MgO < 18 wt.%, distinct to those of komatiites with MgO ≥ 18 wt.%[11,12]; (2) They should have reliable formation ages traced by isotopic dating methods (mainly via whole-rock Sm-Nd, Re-Os and magmatic zircon U-Pb isotopic systematics, which are commonly used to mafic rock dating, and believed to be relatively accurate and not easily susceptible to late thermal events) or limited by geological relationships, for example, the

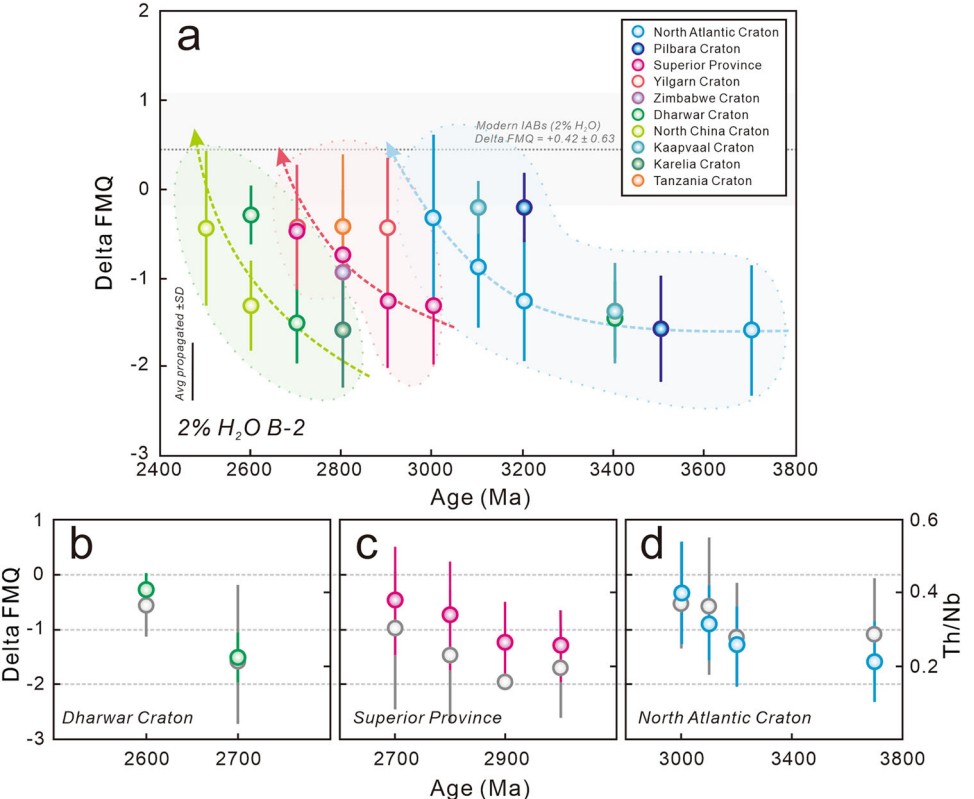

**Fig. 2 Redox state of Archean Basalt-2 (B-2) samples at 2 wt.% H₂O. a,** The $fO_2$ estimations of B-2 samples. The $fO_2$ value and uncertainty (grey interval) of modern island arc basalts (IABs) at 2 wt.% $H_2O$ were used for comparison[14]. The uncertainties in the bottom left corner represent the average 1 standard deviation (SD) propagation of B-2 samples. **b-d** The time evolution of $fO_2$ estimations and Th/Nb ratios in the Dharwar Craton, Superior Province and North Atlantic Craton, respectively. The grey symbols are the B-2 Th/Nb ratios. Error bars show the 1 SD of the means.

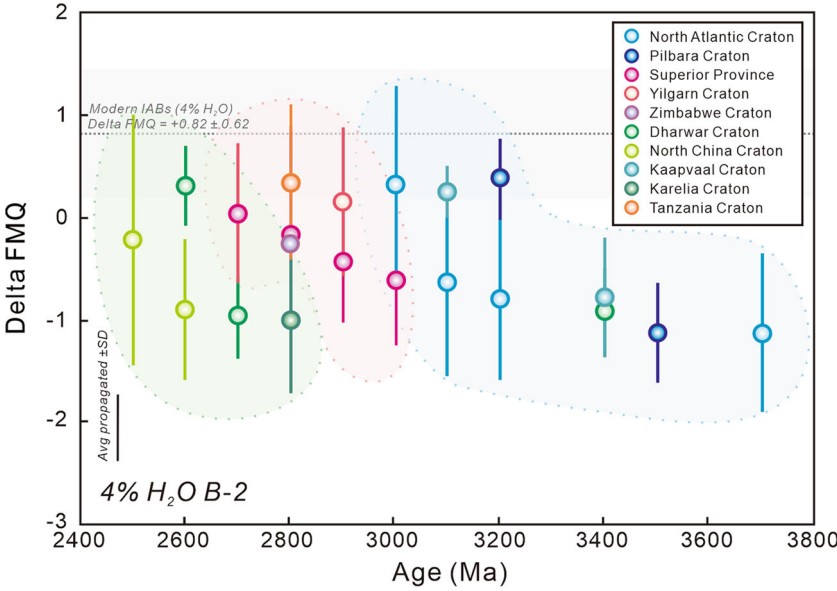

**Fig. 3 Redox state of Archean Basalt-2 (B-2) samples at 4 wt.% H₂O.** The $fO_2$ value and uncertainty (grey interval) of modern island arc basalts (IABs) at 4 wt.% $H_2O$ were used for comparison[14]. Error bars show the 1 standard deviation of the means.

interlayered relationship with felsic volcanic rocks or the intrusive relationship with granitoids and mafic dykes, providing the lower limit to their respective formation age); (3) The least-altered Archean basalts are reported to have low loss on ignition (LOI < 6 wt.%) and negligible Ce anomalies (0.9 < Ce/Ce* < 1.1, calculated as $Ce_N$ / Sqrt($La_N$ × $Pr_N$))[57]; and (4) Basalt samples that have been reported in the literature with significant crustal contamination are excluded. After these processes, this database comprises a total of 2304 basalt samples that cover fourteen Archean cratons (Supplementary Fig. 1; Supplementary Table 1).

**Near fractional melting modeling**. In the process of modelling, the content of trace element $i$ in the instantaneous melt is calculated by $C_{i,m}^{n+1} = \frac{C_{i,r}^n}{D_{i,bulk}^{n+1} + F(1 - D_{i,bulk}^{n+1})}$, where $C_{i,m}^{n+1}$ is the element content in the melt at step $n + 1$, $C_{i,r}^n$ is the element content in the residue after melt extraction at step $n$, $D_{i,bulk}^{n+1}$ is the bulk partition coefficient at step $n + 1$, and $F$ is the melt fraction of each step[14]. Element $i$ content in the aggregated melt at step j is calculated by $C_i^{aggr} = \frac{\sum_1^j C_{i,m}^j M_{melt}^j}{\sum M_{melt}^j}$, where $C_i^{aggr}$ and

$\sum M_{melt}^{j}$ are element $i$ content in the aggregated melt and bulk mass at step $j$, respectively, $C_{i,m}^{j}$ and $M_{melt}^{j}$ are the element $i$ content in the melt and mass at step $j$, respectively[14]. The initial V and Ti contents are set to be 79 ppm and 798 ppm, respectively, according to the depleted mantle (DM) compositions[31]. The initial mineral compositions apply orthopyroxene (Opx) Wo# = 4.38 (Wo# = $X_{Wo}$ / ($X_{Wo} + X_{En} + X_{Fs}$), in which $X_{Wo}$, $X_{En}$ and $X_{Fs}$ are fractions of wollastonite, enstatite and ferrosilite, respectively), clinopyroxene (Cpx) $Al^T$ = 0.17 (Al in the tetrahedron-coordinated site), Opx $Al^T$ = 0.15 and spinel (Sp) Cr# = 10.65 (Cr# = $Cr_2O_3$ / ($Cr_2O_3 + Al_2O_3$) on a molar basis), respectively[30].

**Parameter selection of V-Ti redox proxy.** In model A, the initial mineral assemblages of peridotite constitute by 57% olivine (Ol), 28% Opx, 13% Cpx and 2% Sp[30]. The melting reactions of anhydrous depleted peridotite at 1.0 GPa were used in this model[27]. The melt productivity of 0.23 °C for peridotite and 0.14 °C for harzburgite were used to calculate the melting degrees and temperatures[14]. The recent empirical equations were used to calculate the Kds of V in Ol, Opx, Cpx and Sp, and Ti in Opx and Cpx at each melt fraction[14]. The initial mineral assemblages in model B consists of 52.3% Ol, 17.4% Opx, 26.2% Cpx and 4.1% Grt[28], and the melting reactions were performed by anhydrous peridotite at ~3.0 GPa[28]. In models C and D, the initial mineral assemblages are 53% Ol, 37% Opx, and 9.3% amphibole (Amph), and 52% Ol, 40% Opx, 0.6% Cpx, 7.7% Amph and 0.1% Grt, respectively[29]. The melting reactions were performed on a depleted peridotite, which is metasomatized by a MORB-derived hydrous silicate melt, at low pressure (~1–2 GPa) for modal C and high pressure (~3 GPa) for modal D[29]. The Kds of V in Ol, Opx and Cpx, and Ti in Opx and Cpx at each melt fraction were calculated by the above-mentioned empirical equations with variables of mineral compositions and melting P-T conditions[14]. In addition, the Kds of V in Grt and Amph, and Ti in Ol, Grt and Amph are assumed to be constant and used the values from http://earthref.org/GERM/. All of the calculations are compiled into a macro document and listed in Supplementary Table 2.

## Data availability

The whole-rock geochemical, Nd-isotope and locational data for basalt samples are provided in Supplementary Data 1, and can be also downloaded from the GEOROC database (http://georoc.mpch-mainz.gwdg.de/georoc/). Data for mineral partition coefficients are provided in Supplementary Data 2, and can be obtained from the GERM database (http://earthref.org/GERM/). The authors declare that all data supporting the findings of this study are available online.

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

## Acknowledgements

We sincerely wish to thank C.J. Hawkesworth for constructive suggestions to this contribution. We also want to thank X.S. Li with the help on mathematical statistics and R language technology. This study was financially supported by the National Natural Science Foundation of China (41530207 and 41772188), the China University of Geosciences, Beijing (2-9-2021-023), the Postdoctoral 823 Program for Innovative Talents of Shandong Province (SDBX2021003) and the Australian Research Council (FL160100168).

## Author contributions

L.G., S.W.L., and P.A.C. conceived and designed the project, and wrote the manuscript. J.T.W. helped to calculate the oxygen fugacity. G.Z.S., Y.L.H. and F.Y.H. built the geochemical database and performed data processing. All the authors contributed to the interpretation of the results and the revision of this manuscript.

## Competing interests

The authors declare no competing interests.
