## [Peer Review File · Nature Communications]

REVIEWER COMMENTS

Reviewer #1 (Remarks to the Author):

General comments

Overall this is a very good study and has very important implications for crustal recycling processes and geological interactions between the mantle and surface environments in the Archean. The study deals with the changes in trace element compositions of intermediate to ultramafic volcanic rocks derived from Archean mantle and investigates the changes in oxygen fugacity in the mantle between 3.8 and 2.5 billion years ago. These are very important and hot topics in Earth sciences. The authors compiled data from a large number samples collected from all major cratons, covering most of the Archean geological record, resulting in statistically reliable values on oxygen fugacity in the mantle and representative trace element ratios (e.g., Nb/La, La/Yb) in mantle-derived volcanic rocks. The authors did a very good job in screening the geochemical data for alteration and crustal contamination to determine the reliable samples. In addition, they provide a good summary of the geochemical characteristics of the classified volcanic rocks (e.g., Fig. 3).

The authors present a good introduction to the study topics. Objectives are clear. Methods that are used in the manuscript to perform the calculation and classify the volcanic rocks are valid and reliable. Geodynamic and geochemical arguments presented in the manuscript generally agree with the data reported in the manuscript. The topic of the manuscript is of great interest to the international geological community. The manuscript is timely as well as innovative. Figures and tables are all clear and necessary to convey the information. The references are appropriate and adequate. As outlined below, there are, however, a few issues that the authors should address to improve the manuscript. Accordingly, I recommend the acceptance of the manuscript with moderate revisions. I hope that my comments are of help in the revision process.

Specific Comments

1. Line 50: It would be better to replace 'oxidized' by 'elevated' or 'increased'.
2. Line 70: Do you mean 'sensitive than' by 'sensitive to'?
3. Line 122: Replace 'distinct to' by 'distinct from'.
4. Line 158: Replace 'methodology' by 'method' because methodology means the study of methods.
5. Lines 171-172: This statement is inconsistent with field geology, igneous and metamorphic petrology (please Polat and Hofmann, 2003; Nutman et al., 2009; Turner et al., 2014; Nutman et al., 2020, 2021; Nakamura et al., 2020; Garde et al., 2020; Guotana et al., 2021). You need to explain why your interpretation contradicts the geological record.
6. Lines 163-180: I agree that komatiites are more abundant in the Archean rock record; however, none of the known Archean greenstone belts contains more than 15% komatiites. Most greenstone belts are dominated by basalts, and komatiites are only <10% of the exposed volcanic rocks. Can you provide an example of an Archean greenstone belt that contains >40 % komatiite? 48% komatiite database is likely reflect bias sampling and analyses of komatiites. This issue needs to be addressed.
7. Line 352: Replace '15.0' by '15'.

References

Garde, A.A., Windley, B.F., Kokfelt, T.F., Keulen, N., 2020. Archean plate tectonics in the North Atlantic Craton of West Greenland revealed by well-exposed horizontal crustal tectonics, island arcs and tonalite-trondhjemite-granodiorite complexes. *Frontiers in Earth Science* 8, 540997. doi: 10.3389/feart.2020.540997.

Guotana, J.M., Morishita, T., Nishio, I., Tamura, A., Mizukami, T., Tani, K., Harigane, Y., Szilas, K., Pearson, D.G., 2021. Deserpentinization and high-pressure (eclogite-facies) metamorphic features in the Eoarchean ultramafic body from Isua, Greenland, *Geoscience Frontiers*. doi: <https://doi.org/10.1016/j.gsf.2021.101298>.

Nakamura, H., Sano, A., Kagami, S., Yokoyama, T., Ishikawa, A., Komiya, T., Iwamori, H.,

2020. Compositional heterogeneity of Archean mantle estimated from Sr and Nd isotopic systematics of basaltic rocks from North Pole, Australia, and the Isua supracrustal belt, Greenland. *Precambrian Research* 347, 105803.

Nutman, A.P., Friend, C.R.L., Paxton, S., 2009. Detrital zircon sedimentary provenance ages for the Eoarchean Isua supracrustal belt southern West Greenland: Juxtaposition of a ca. 3700 Ma juvenile arc assemblage against an older complex with 3920-3800 Ma components. *Precambrian Research* 172, 212-233.

Nutman, A.P., Bennett, V.C., Friend, C.R.L., Keewook Yi, 2020. Eoarchean contrasting ultra-high-pressure to low-pressure metamorphisms (< 250 to >1000 °C/GPa) explained by tectonic plate convergence in deep time. *Precambrian Research* 344, 105770.

Nutman, A.P., Scicchitano, M., Friend, C.R.L., Bennett, V.C., Chivas, A.R., 2021. Isua (Greenland) ~3700 Ma Ultra High Pressure mantle meta-peridotite olivine Mg# and $\delta^{18}O$ signatures show connection between the early mantle and hydrosphere: Geodynamic implications. *Precambrian Research*, 361, 106249. doi.org/10.1016/j.precamres.2021.106249.

Polat, A., Hofmann, A.W., 2003. Alteration and geochemical patterns in the 3.7-3.8 Ga Isua greenstone belt, West Greenland. *Precambrian Research* 126, 197-218.

Turner, S., Rushmer, T., Reagan, M., Moyen, J.-F., 2014. Heading down early on? Start of subduction on Earth. *Geology* 42, 139-142.

Reviewer #2 (Remarks to the Author):

Review of Gao et al. "Secular oxidation of Archean upper mantle caused by increasing crustal recycling"

The paper by Gao et al is a short paper that applies a V-Ti proxy to Archean basalts from the Georoc database to establish if there are any secular trends in Archean mantle fO_2 . Based on application of the trace element redox proxy, the authors believe that a secular oxidation is observed in both tholeiitic ("MORB") and calc-alkaline ("IAB") basalts, in addition to the Archean mantle being approximately 1 log unit lower than present day MORB-source mantle. The paper presents an interesting (if not quite original) viewpoint – to establish the presence/absence of secular trends in mantle fO_2 . This has been conducted previously using a V-Sc proxy (e.g. Lee, Aulbach) on basalts (assuming basalts adequately represent their upper mantle sources), work on komatiites (e.g. Nicklas), as well as studies of mantle xenoliths (ecogites representing basalts: Smart, Aulbach, Burness; also peridotites: Woodland, Creighton, Tappe). While this new contribution apparently does see firstly a generally more reduced Archean mantle, it also claims to see a secular increase in mantle fO_2 from 3.2 to 2.5 Ga. In addition, the paper does some statistical analysis on the Georoc database to comment on the proportion of mafic volcanic rocks in the Archean and uses their geochemistry to approximate magma genesis. Papers such as this one are always beneficial – taking the large quantities of geochemical data that exist and try and work on "big picture problems" – in this case, the evolution (or not) of the oxidation state of Earth's upper mantle – which is timely and topical currently in the geosciences.

However, there are several drawbacks to this study that prevent immediate publication. Firstly, the paper is extremely short – my guess this paper was prepared for *Nature / Nature Geoscience*. In *Communication*, longer papers are allowed and this paper would benefit at every step from more information on techniques, assumptions, discussion etc. Secondly – the authors somewhat misrepresent their B-1 fO_2 data – they claim to see a clear increase in fO_2 from 3.2 – 2.7 Ga; From Figure 2a, I cannot see this same clear trend. I see a very heterogeneous distribution of data with no clear trends. Thus, from the basalts that are thought to sample the upper mantle (equivalent to modern DM I think), there is no clear secularity. Thirdly, it is not clear at all the uncertainties on the fO_2 determinations. The uncertainties are very low (~0.1 log units) and it must be established how the uncertainties are determined, especially since multiple types of

data (bulk trace element data, P-T determinations, melt modelling) and assumptions are being integrated into the final redox composition. Finally, the last section of the paper is in somewhat stark contrast to the rest of the paper. Instead of more fully discussing the redox potential of Eo- to Neoproterozoic mantle, or the sources of oxidation or comparison to mantle xenolith work, there is a poorly explained statistical breakdown of the Georoc database and some further geochemical modelling about the magma sources in the B-1 and B-2 basalts. It just does not seem to naturally flow or fit with this paper. If the authors want to integrate geochemical modelling of mantle magma sources into their paper, why not link to the redox results with modelling of incompatible elements with fO_2 , or other redox proxies? I think developing and exploring the findings in a longer format paper would be extremely beneficial and make for a very interesting contribution to mantle geochemistry.

21-23: In the opening line, "Earth differentiation" and "nature of chemical equilibration between mantle, ocean and atmosphere" seem to be essentially saying the same thing?

44: Representation of references: Perhaps Nicklas et al., (2019) found a more significant trend in ΔFMQ vs. time, but certainly Nicklas et al., (2018) did not find a significant trend.

47: I think there might be a clarity issue here. While Sossi (and others) might argue that the thermal perturbation (i.e. mantle plume) originates at the core-mantle or perhaps more plausibly at the TZ, which does not mean that the komatiite melt itself originates from such great depths. Rather, upon decompression in the upper mantle, the komatiite forms closer to $\sim 6 - 9$ GPa (or so).

49: The justification for using basalts over komatiites is somewhat unbalanced. Using basalts as a proxy for Archean mantle is also subject to uncertainty. For instance, although one can attempt to use geochemical proxies to determine magmatic setting, the elemental, isotopic and even redox heterogeneity of modern MORB complicates using geochemical parameters to infer the tectonic setting of magma genesis. This will crop up again at line 93-ish where the Nb/La proxy is used to evaluate subduction zone vs MORB-type basalts. I also see in the supplement that while low La/ Yb is generally characteristic for MORB-type tholeiites and most B-1 basalts follow this trend, calc-alkaline basalts (and many B-2 basalts) can also have low "tholeiite-like" La/Yb. So I wonder how robust using these geochemical proxies for tectono-magmatic setting is...

Additionally, there may be issues of basalts preserving "pristine" upper mantle redox, because degassing and S might play a role in modifying the basalt fO_2 . There have been instances of finding correlations between redox-sensitive elements and S in OIBs (work of Kelley and Cottrell), so the problem of degassing and its effects on redox is problematic for basalts (which are generated through both lower Ts and increments of melting vs. komatiites). Of course, using komatiites in redox proxies may also have drawbacks, but the authors should offer a somewhat more balanced evaluation of the justification/methodology here.

70: perhaps more sensitive "than" instead of "to"

114: reference for the software?

124: What is lower "level" conditions? I would argue the "lower level" P-T for B-2 are not significantly lower at all - in fact the P-T ranges overlap completely, although B-1 does extend to higher T and P.

136 - 155-ish: This section is contradictory and does not accurately represent the results reported on Fig2. I would agree with the authors that in general, the basalts (if they accurately represent their mantle sources) demonstrate the upper mantle has a heterogeneous fO_2 in the Archean, which appears on average to be reduced by about 1 log unit compared to modern MORB. Basalts from some locations (e.g. Dharwar) appear to show a "progressive" oxidation, but then other locations show opposite trends: that is progressive reduction with younging (Superior). In summary, the B-1 basalts, which are presumed to represent the upper mantle, do not clearly show a temporal evaluation

of mantle fO_2 and I believe that a statistical analysis of the data points in Fig2a would not show any clear trends. The secular oxidation as represented by the B-2 samples is much more apparent – but this is probably because they are derived from heterogeneous sources that may have experienced progressively more input of recycled oxidized sources.

The uncertainties in the determined fO_2 values are very small – on the order of 0.1 log units. Oxybarometry for mantle-derived rocks generally gives uncertainties at minimum 0.5 log units. In the supplement, there is no discussion regarding the calculation of uncertainties on the fO_2 calculation. Certainly sources of error that arise from the P-T calculations and trace element analyses are propagated into the overall redox composition calculation? A rigorous outline of the uncertainties should be presented, especially since the total variation of the B-1 and B-2 basalts occurs within <2 log units, so some of the “trends” might disappear if the fO_2 uncertainties were to increase.

Also pertaining to the calculation of the fO_2 – the melt modelling equations are conducted on garnet and spinel lherzolites – from “primitive mantle”. For the Eoarchean basalts this seems appropriate, but some of the younger basalts that formed after production of more voluminous continental crust (towards the end of the Archean) might be better modelled using a “depleted mantle” type peridotite.

“Increase in crustal recycling” section: This section comes as a sharp contrast to the rest of the paper. While more “space” could have been used to fully evaluate the variability of upper mantle fO_2 with geochemical (or even isotopic) proxies to tease out the type and proportion of crustal contamination or mantle reservoir. Instead the paper turns to a statistical breakdown of the GEOROC Archean mafic rock database and some geochemical discrimination diagrams. Rather large conclusions are made from this statistical breakdown: “non subduction geodynamic regime play a dominant role” from 3.8 to 3.2 Ga based on % of basalt and komatiite. Firstly, the proportion of komatiite seems very high – it is well known from greenstone belts on cratons that the proportion of komatiite is quite small. How were the proportions (e.g. 48% for komatiite between 3.2 – 3.8 Ga) calculated? Surely not from a simple % from the GEOROC database entries? I did not see any information in the Methods or Supplement. This needs to be fully discussed because the komatiite #s seem very high. Secondly, such statements about the operation of subduction in the Archean need to be better supported, using perhaps complimentary evidence from other studies that have further investigated Archean rocks for “signs” of subduction tectonics. Take your pick here: Shirey, Dhuime, Kamber, etc.

Then we come to Figure 3d-f which models basalt source geochemistry based on some trace element ratios. This type of figure and argument have been made by many papers in the past, and it is not clear how or why this fits with the theme of secular oxidation proposed by the title and introduction of this paper. Obviously the more geochemically enriched B-2 basalts tend towards more “crustal” inputs...it would have been more interesting to do such modelling in terms of redox, considering what we know of Archean recycled oceanic crust as represented by eclogite xenolith studies by Aulbach et al (op. cit) and Smart et al. 2017, 2021a,b.

Reviewer #3 (Remarks to the Author):

This contribution reports the redox evolution of Archean mantle mainly based on the mantle-derived volcanic rocks (Archean basalts) on Earth. Their updated redox proxies suggest that the Archean mantle was undergoing a secular oxidation process, owing to increased crustal recycling via plate subduction. This study adds to our understanding of Earth's early evolution by providing insight into the elemental evolution of the early Precambrian mantle.

I still have reservations regarding the derivation from the raw data to the conclusions, hence I suggest a minor revision of this work before publishing, taking into account the issues listed below.

(1) This discovery is based primarily on data from Earth's Archean basalts. My first concern is how basalt ages were determined. Are the ages accurate, given that dating basic volcanic rocks is typically difficult?

(2) According to the authors, the oxidation of the Archean upper mantle is due to increased crustal recycling into the mantle, potentially due to plate subduction. While early subduction tectonics on Earth is a hotly debated topic. Is there another mechanism that accounts for the recycling of crustal materials or the oxidation of the mantle? It's possible that more discussion of various options is required.

(3) My final question concerns the redox state of the Archean earth's surface. Was it enough to alter the redox status of the mantle, given that Earth's great oxidation occurred much later?

Wenjiao Xiao

Response to comments of Reviewer #1

Overall this is a very good study and has very important implications for crustal recycling processes and geological interactions between the mantle and surface environments in the Archean. The study deals with the changes in trace element compositions of intermediate to ultramafic volcanic rocks derived from Archean mantle and investigates the changes in oxygen fugacity in the mantle between 3.8 and 2.5 billion years ago. These are very important and hot topics in Earth sciences. The authors compiled data from a large number samples collected from all major cratons, covering most of the Archean geological record, resulting in statistically reliable values on oxygen fugacity in the mantle and representative trace element ratios (e.g., Nb/La, La/Yb) in mantle-derived volcanic rocks. The authors did a very good job in screening the geochemical data for alteration and crustal contamination to determine

the reliable samples. In addition, they provide a good summary of the geochemical characteristics of the classified volcanic rocks (e.g., Fig. 3).

The authors present a good introduction to the study topics. Objectives are clear. Methods that are used in the manuscript to perform the calculation and classify the volcanic rocks are valid and reliable. Geodynamic and geochemical arguments presented in the manuscript generally agree with the data reported in the manuscript. The topic of the manuscript is of great interest to the international geological community. The manuscript is timely as well as innovative. Figures and tables are all clear and necessary to convey the information. The references are appropriate and adequate. As outlined below, there are, however, a few issues that the authors should address to improve the manuscript. Accordingly, I recommend the acceptance of the manuscript with moderate revisions. I hope that my comments are of help in the revision process.

1. Line 50: It would be better to replace ‘oxidized’ by ‘elevated’ or ‘increased’.

Revised. See line 45.

2. Line 70: Do you mean ‘sensitive than’ by ‘sensitive to’?

Revised. See line 73.

3. Line 122: Replace ‘distinct to’ by ‘distinct from’.

Revised. See lines 146-147.

4. Line 158: Replace ‘methodology’ by ‘method’ because methodology means the study of methods.

Revised. See line 155.

5. Lines 171-172: This statement is inconsistent with field geology, igneous and metamorphic petrology (please Polat and Hofmann, 2003; Nutman et al., 2009; Turner et al., 2014; Nutman et al., 2020, 2021; Nakamura et al., 2020; Garde et al., 2020; Guotana et al., 2021). You need to explain why your interpretation contradicts the geological record.

References

Garde, A.A., Windley, B.F., Kokfelt, T.F., Keulen, N., 2020. Archaean plate tectonics in the North Atlantic Craton of West Greenland revealed by well-exposed horizontal crustal tectonics, island arcs and tonalite-trondhjemite-granodiorite complexes. *Frontiers in Earth Science* 8, 540997. doi: 10.3389/feart.2020.540997.

Guotana, J.M., Morishita, T., Nishio, I., Tamura, A., Mizukami, T., Tani, K., Harigane, Y., Szilas, K., Pearson, D.G., 2021. Deserpentinization and high-pressure (eclogite-facies) metamorphic features in the Eoarchean ultramafic body from Isua, Greenland, *Geoscience Frontiers*. doi:<https://doi.org/10.1016/j.gsf.2021.101298>.

Nakamura, H., Sano, A., Kagami, S., Yokoyama, T., Ishikawa, A., Komiya, T., Iwamori, H., 2020. Compositional heterogeneity of Archean mantle estimated from Sr and Nd isotopic systematics of basaltic rocks from North Pole, Australia, and the Isua supracrustal belt, Greenland. *Precambrian Research* 347, 105803.

Nutman, A.P., Friend, C.R.L., Paxton, S., 2009. Detrital zircon sedimentary provenance ages for the Eoarchean Isua supracrustal belt southern West Greenland: Juxtaposition of a ca. 3700 Ma juvenile arc assemblage against an older complex with 3920-3800 Ma components. *Precambrian Research* 172, 212-233.

Nutman, A.P., Bennett, V.C., Friend, C.R.L., Keewook Yi, 2020. Eoarchean contrasting ultra-high-pressure to low-pressure metamorphisms (< 250 to >1000 °C/GPa) explained by tectonic plate convergence in deep time. *Precambrian Research* 344, 105770.

Nutman, A.P., Scicchitano, M., Friend, C.R.L., Bennett, V.C., Chivas, A.R., 2021. Isua (Greenland) ~3700 Ma Ultra High Pressure mantle meta-peridotite olivine Mg# and $\delta^{18}\text{O}$ signatures show connection between the early mantle and hydrosphere: Geodynamic implications. *Precambrian Research*, 361, 106249.

doi.org/10.1016/j.precamres.2021.106249.

Polat, A., Hofmann, A.W., 2003. Alteration and geochemical patterns in the 3.7-3.8 Ga Isua greenstone belt, West Greenland. *Precambrian Research* 126, 197-218.

Turner, S., Rushmer, T., Reagan, M., Moyen, J.-F., 2014. Heading down early on? Start of subduction on Earth. *Geology* 42, 139–142.

Thanks for your suggestion. These references are very helpful, and some of them have been added to appropriate places in the updated text. We agree that the plate subduction or similar process could have occurred in the North Atlantic Craton during ~3.8-3.2 Ga based on previous studies on the petrology, structural geology and geochemistry. In this study, we found that the North Atlantic Craton preserves the earliest evolutionary signatures in the world and the close relationship between the mantle redox state and basalt Th/Nb ratios (Figures 1 and 2), indicating that the early crustal recycling, which may be processed by plate subduction or lithospheric drips, caused the gradual increase of mantle oxygen fugacity.

6. Lines 163-180: I agree that komatiites are more abundant in the Archean rock record; however, none of the known Archean greenstone belts contains more than 15% komatiites. Most greenstone belts are dominated by basalts, and komatiites are only <10% of the exposed volcanic rocks. Can you provide an example of an Archean greenstone belt that contains >40 % komatiite? 48% komatiite database is likely reflect bias sampling and analyses of komatiites. This issue needs to be addressed.

Thank you for this comment. In the last version, we did overestimate of the proportion of komatiites in the worldwide greenstone belts without full consideration of bias sampling. Reviewer #2 said that the statistical breakdown of the proportions of different rock types is in somewhat stark contrast to the rest of the paper, and suggested that the main part of this manuscript should focus on the redox state of Archean mantle and the origin of oxidation. Therefore, the related contents have been deleted. The discussion in the updated text has focused on the redox state of Archean

ambient and modified mantle (lines 174-191), the fO_2 variations beneath Archean cratons and temporal craton groups during Eo- to Neoproterozoic and their implications (lines 195-234), the origin of mantle oxidation (lines 236-270), and the possible geodynamic mechanism (lines 271-276).

7. Line 352: Replace '15.0' by '15'.

Revised.

Based on your and other reviewers' comments here and other previous studies, the komatiites are generally less than 10% of the exposed Archean volcanic rocks in the global range, and we consider that the source materials of komatiitic magma are complicated. The sources might come from various mantle levels with materials from either the core-mantle or the upper-lower mantle boundaries. Furthermore, komatiites assimilate ambient mantle materials and oceanic crust residues during upwelling of mantle plume source. Therefore, the redox state signed by komatiites might not effectively reflect the nature of upper mantle. The updated text should focus on the redox state of Archean basalts rather than that of komatiites, therefore, the related descriptions of komatiites in the last version of this article has been depleted; See lines 46-51.

Response to comments of Reviewer #2

The paper by Gao et al is a short paper that applies a V-Ti proxy to Archean basalts from the Georoc database to establish if there are any secular trends in Archean mantle fO_2 . Based on application of the trace element redox proxy, the authors believe that a secular oxidation is observed in both tholeiitic ("MORB") and calc-alkaline ("IAB") basalts, in addition to the Archean mantle being approximately 1 log unit lower than present day MORB-source mantle. The paper presents an interesting (if not quite original) viewpoint – to establish the presence/absence of secular trends in mantle fO_2 . This has been conducted previously using a V-Sc proxy

(e.g. Lee, Aulbach) on basalts (assuming basalts adequately represent their upper mantle sources), work on komatiites (e.g. Nicklas), as well as studies of mantle xenoliths (ecogites representing basalts: Smart, Aulbach, Burness; also peridotites: Woodland, Creighton, Tappe). While this new contribution apparently does see firstly a generally more reduced Archean mantle, it also claims to see a secular increase in mantle fO_2 from 3.2 to 2.5 Ga. In addition, the paper does some statistical analysis on the Georoc database to comment on the proportion of mafic volcanic rocks in the Archean and uses their geochemistry to approximate magma genesis. Papers such as this one are always beneficial – taking the large quantities of geochemical data that exist and try and work on “big picture problems” – in this case, the evolution (or not) of the oxidation state of Earth’s upper mantle – which is timely and topical currently in the geosciences.

Thank you for reviewing our manuscript.

Previous studies determined the redox state of the Archean upper mantle mainly from: (1) The $Fe^{3+}/\Sigma Fe$ ratio in pristine mantle and mantle-derived rocks (Woodland and Koch, 2003; Aulbach et al., 2019); (2) The V behavior in komatiites (Nicklas et al., 2018, 2019); and (3) The V-Sc redox proxy in MORB-like basalts (Anser and Lee, 2004; Aulbach and Stagno, 2016). These methods provide an in principle estimate of oxygen fugacity of the upper mantle. However, in term of the most Archean samples, the $Fe^{3+}/\Sigma Fe$ ratios of minerals and glasses may be readily modified during alteration, metamorphism and syn-eruptive process (Nicklas et al., 2018, 2019). And the sources of komatiitic magma are complicated, and might come from the various mantle levels with transporting materials from either the core-mantle or the upper-lower mantle boundaries. Furthermore, komatiites assimilate ambient mantle materials and oceanic crust residues during upwelling of mantle plume source. Therefore, the redox state signed by komatiites might not effectively reflect the nature of upper mantle. Compared to komatiites, the widespread appearance of Archean basalts offers alternative insights into mantle redox state of the early earth (Chen et al., 2020). In our recent study, we found the V-Ti systematics is superior to previous V-Sc pair

because Sc content of the melts are easily influenced by garnet residues (see Wang et al., 2019). Furthermore, our updated V-Ti redox proxy has fully considered the influence of different mantle melting reactions and the relationships among the partition coefficients of V and Ti in mantle minerals, the melting P-T conditions, the water contents and the compositions of mantle minerals (Wang et al., 2019), of which influences were not fully introduced by previous studies. Therefore, we believe that the V-Ti systematics is a more reliable and accurate method estimating the redox state of Archean upper mantle (lines 37-63 and 72-75).

Cited references:

Aulbach, S. et al. Evidence for a dominantly reducing Archean ambient mantle from two redox proxies, and low oxygen fugacity of deeply subducted oceanic crust. *Sci. Rep.* 9, 20190 (2019).

Woodland, A.B. & Koch, M. Variation in oxygen fugacity with depth in the upper mantle beneath the Kaapvaal craton, Southern Africa. *Earth Planet. Sci. Lett.* 214, 295-310 (2003).

Smart, K.A., Tappe, S., Simonetti, A., Simonetti, S.S., Woodland, A.B. & Harris, C. Tectonic significance and redox state of Paleoproterozoic eclogite and pyroxenite components in the Slave cratonic mantle lithosphere, Voyageur kimberlite, Arctic Canada. *Chem. Geol.* 455, 98-119 (2017).

Smart, K. A. et al. Metasomatized eclogite xenoliths from the central Kaapvaal craton as probes of a seismic mid-lithospheric discontinuity. *Chemical Geology* 578, 120286 (2021).

Nicklas, R. W. et al. Secular mantle oxidation across the Archean-Proterozoic boundary: Evidence from V partitioning in komatiites and picrites. *Geochim. Cosmochim. Acta* 250, 49-75 (2019).

Nicklas, R. W., Puchtel, I. S. & Ash, R. D. Redox state of the Archean mantle Evidence from V partitioning in 3.5-2.4 Ga komatiites. *Geochim. Cosmochim. Acta* 222, 447-466 (2018).

Anser Li, Z. X. & Aeolus Lee, C. T. The constancy of upper mantle fO_2 through time

inferred from V/Sc ratios in basalts. *Earth Planet. Sci. Lett* 228, 483-493 (2004).

Aulbach, S. & Stagno, V. Evidence for a reducing Archean ambient mantle and its effects on the carbon cycle. *Geology* 44, 751-754 (2016).

Chen, K. et al. How mafic was the Archean upper continental crust? Insights from Cu and Ag in ancient glacial diamictites. *Geochim. Cosmochim. Acta* 278, 16-29 (2020).

Wang, J. et al. Oxidation State of Arc Mantle Revealed by Partitioning of V, Sc, and Ti Between Mantle Minerals and Basaltic Melts. *J. Geophys. Res. Solid Earth* 124, 4617-4638 (2019).

However, there are several drawbacks to this study that prevent immediate publication.

1. Firstly, the paper is extremely short – my guess this paper was prepared for *Nature* / *Nature Geoscience*. In *Communication*, longer papers are allowed and this paper would benefit at every step from more information on techniques, assumptions, discussion etc.

Revised.

Thanks for this comment.

Based on the suggestion here, we have enhanced the description of the calculation method and result of oxygen fugacity, the uncertainties and propagations and the final discussion. See lines 299-332, 174-234, 156-173 and 193-276. The total length of the main text has been increased from less than 2000 to 3200 words in the updated version.

2. Secondly, the authors somewhat misrepresent their B-1 fO₂ data – they claim to see a clear increase in fO₂ from 3.2 – 2.7 Ga; From Figure 2a, I cannot see this same clear trend. I see a very heterogeneous distribution of data with no clear trends. Thus, from the basalts that are thought to sample the upper mantle (equivalent to modern DM I think), there is no clear seculariry.

Good point.

Revised.

We have made the following modifications: (1) Expanded the Archean basalt database to include the recent published 172 basalt data; (2) Modified our models by using melting reactions of the depleted peridotite instead of those originally melting reactions of fertile peridotite. See more details in the reply to comment 15; (3) Chosen basalts with MgO \geq 8.5 wt.% to calculate the mantle oxygen fugacity and avert possible effects from the late overprint of fractional crystallization and crust contamination (Lee et al., 2009). After these modifications, the results are more reliable and have significantly changed (Figs. 1-3). Beneath most Archean cratons or the three temporal craton groups, the oxygen fugacity of both ambient and modified mantle exhibits \sim 1.0 log unit increase. For example, calculated fO_2 values of modified and ambient mantle beneath the North Atlantic Craton increase from $\Delta FMQ -1.58 \pm 0.74$ (1SD) to $\Delta FMQ -0.33 \pm 0.93$ at 2 wt.% H₂O and $\Delta FMQ -1.64 \pm 0.40$ to $\Delta FMQ -0.56 \pm 0.16$ between \sim 3.8 and 3.0 Ga, respectively; the Superior Province from $\Delta FMQ -1.31 \pm 0.66$ to $\Delta FMQ -0.47 \pm 0.54$ at 2 wt.% H₂O and $\Delta FMQ -1.79 \pm 0.48$ to $\Delta FMQ -1.16 \pm 0.52$ between \sim 3.0 and 2.7 Ga, respectively; the North China Craton from $\Delta FMQ -1.31 \pm 0.51$ to $\Delta FMQ -0.44 \pm 0.87$ at 2 wt.% H₂O and $\Delta FMQ -2.14 \pm 0.09$ to $\Delta FMQ -1.09 \pm 0.25$ between \sim 2.8 and 2.5 Ga, respectively (Figs. 1,2). Similar fO_2 tendencies are also shown beneath the Pilbara, Kaapvaal, Yilgarn and Dharwar cratons. Based on similarities and differences in the temporal evolution and the mantle fO_2 tendencies of the cratons, they can be divided into three temporal groups: (1) The \sim 3.8-3.0 Ga craton group (the North Atlantic, Kaapvaal and Pilbara cratons); (2) The \sim 3.0-2.7 Ga craton group (the Superior Province, the Yilgarn, Tanzania and Zimbabwe cratons); and (3) The \sim 2.8-2.5 Ga craton group (the North China and Dharwar cratons). In addition, the nearly constant starting fO_2 values of each craton suggest that a redox homogeneous mantle prior to \sim 3.8 Ga. Furthermore, the final fO_2 values are also nearly constant, indicating that the mantle redox state may be an effective indicator of craton maturation and cratonization. See the updated Figures 1-3 and lines 195-234.

Cited references:

Lee, C.-T. A., Luffi, P., Plank, T., Dalton, H. & Leeman, W. P. Constraints on the depths and temperatures of basaltic magma generation on Earth and other terrestrial planets using new thermobarometers for mafic magmas. *Earth Planet. Sci. Lett.* 279, 20-33 (2009).

3. Thirdly, it is not clear at all the uncertainties on the fO_2 determinations. The uncertainties are very low (~ 0.1 log units) and it must be established how the uncertainties are determined, especially since multiple types of data (bulk trace element data, P-T determinations, melt modelling) and assumptions are being integrated into the final redox composition.

Revised.

Sources of uncertainties that arise from trace element analyses, mantle compositional heterogeneity, P-T determinations and systematic biases for the Kds of V in mantle minerals and the fO_2 are incorporated into the application of V-Ti redox proxy and conversion of V/Ti ratio into the mantle fO_2 . These uncertainties are incorporated as the following: (1) Analytical uncertainties of V and Ti are set to 5 %, assuming that all of samples collected in this study were analyzed by the inductively-coupled plasma mass-spectrometry (ICP-MS) technique (Nicklas et al., 2019; Wang et al., 2019); (2) Uncertainties of mantle heterogeneity were taken as 7 % for V and 12 % for Ti (Wang et al., 2019); (3) Uncertainties of P-T determinations are based on an absolute value of 0.20 GPa and 3 %, respectively (Lee et al., 2009), assuming that there were no additional uncertainties in the water content assessment; (4) Estimate of uncertainties and systematic biases for the functions between Kds of V in mantle minerals and the fO_2 were incorporated into further propagations (Wang et al., 2019). In this study, uncertainties (1) and (2) were firstly propagated into the calculation of the compositions of V and V/Ti in the melts and their Kds in bulk rock compositions, which were then propagated to the Kds in mantle minerals and fO_2 on

the basis of a full account of uncertainties (3) and (4). On this basis, the average propagated 1 SD uncertainties of the mantle fO_2 revealed by B-1 and -2 samples are 0.40 and 0.32 log units, respectively. See the updated Figures 1-3 and lines 156-173.

Cited references:

Nicklas, R. W. et al. Secular mantle oxidation across the Archean-Proterozoic boundary: Evidence from V partitioning in komatiites and picrites. *Geochim. Cosmochim. Acta* 250, 49-75 (2019).

Wang, J. et al. Oxidation State of Arc Mantle Revealed by Partitioning of V, Sc, and Ti Between Mantle Minerals and Basaltic Melts. *J. Geophys. Res. Solid Earth* 124, 4617-4638 (2019).

Lee, C.-T. A., Luffi, P., Plank, T., Dalton, H. & Leeman, W. P. Constraints on the depths and temperatures of basaltic magma generation on Earth and other terrestrial planets using new thermobarometers for mafic magmas. *Earth Planet. Sci. Lett.* 279, 20-33 (2009).

4. Finally, the last section of the paper is in somewhat stark contrast to the rest of the paper. Instead of more fully discussing the redox potential of Eo- to Neoproterozoic mantle, or the sources of oxidation or comparison to mantle xenolith work, there is a poorly explained statistical breakdown of the Georoc database and some further geochemical modelling about the magma sources in the B-1 and B-2 basalts. It just does not seem to naturally flow or fit with this paper. If the authors want to integrate geochemical modelling of mantle magma sources into their paper, why not link to the redox results with modelling of incompatible elements with fO_2 , or other redox proxies? I think developing and exploring the findings in a longer format paper would be extremely beneficial and make for a very interesting contribution to mantle geochemistry.

Thank you for the comment.

Revised.

We have made substantial revision to our discussion section: (1) According to your advice, the original discussion of rock type statistics and geochemical modelling on the sources of B-1 and B-2 samples have been deleted; (2) The updated text has now focused on the redox state of Archean ambient and modified mantle (see lines 174-191), the fO_2 variations and implications beneath Archean cratons and temporal craton groups during Eo- to Neoproterozoic (see lines 195-234), the sources and mechanisms of mantle oxidation (see lines 236-270), and the potential geodynamic regime (see lines 271-276); (3) Your final suggestion of geochemical modelling in terms of redox with incompatible elements and eclogitic xenolith studies is suggestive. But in current situation, there may be some difficulties in model selection, parameter determination and data filtration. We agree that these ideas are better to describe in another longer format paper as you suggested, exploring the mantle geochemical evolution of the early Earth. We will do this research in the near future.

5. 21-23: In the opening line, “Earth differentiation” and “nature of chemical equilibration between mantle, ocean and atmosphere” seem to be essentially saying the same thing?

Revised. The expression ‘Earth differentiation’ is deleted. See lines 22-24.

6. 44: Representation of references: Perhaps Nicklas et al., (2019) found a more significant trend in ΔFMQ vs. time, but certainly Nicklas et al., (2018) did not find a significant trend.

Revised. We have removed the incorrect reference. See lines 44-46.

7. 47: I think there might be a clarity issue here. While Sossi (and others) might argue that the thermal perturbation (i.e. mantle plume) originates at the core-mantle or perhaps more plausibly at the TZ, which does not mean that the komatiite melt itself originates from such great depths. Rather, upon decompression in the upper mantle,

the komatiite forms closer to $\sim 6 - 9$ GPa (or so).

Yes, we agree with the recognition that the komatiite magmas might come from the decompression melting of mantle plume at upper mantle depths. However, based on Fe and Hf isotopes, the magmatic sources of mantle plume might carry, or mix, variable degrees of materials from the core-mantle boundary during upwelling (Nebel et al., 2014). Compared to the dominant role of basalts in the Archean crust (Chen et al., 2020), the komatiites occupy $< 10\%$ of the exposed greenstone belts and the komatiite magmas might be derived from the head of mantle upwelling involving complex mantle materials at different depth levels. Therefore, using komatiites to reflect the nature of the upper mantle might not be suitable. See lines 46-51.

Cited references:

Nebel, O., Campbell, I. H., Sossi, P. A. & Van Kranendonk, M. J. Hafnium and iron isotopes in early Archean komatiites record a plume-driven convection cycle in the Hadean Earth. *Earth Planet. Sci.* 397, 111-120 (2014).

Chen, K. et al. How mafic was the Archean upper continental crust? Insights from Cu and Ag in ancient glacial diamictites. *Geochim. Cosmochim. Acta* 278, 16-29 (2020).

8. 49: The justification for using basalts over komatiites is somewhat unbalanced. Using basalts as a proxy for Archean mantle is also subject to uncertainty. For instance, although one can attempt to use geochemical proxies to determine magmatic setting, the elemental, isotopic and even redox heterogeneity of modern MORB complicates using geochemical parameters to infer the tectonic setting of magma genesis. This will crop up again at line 93-ish where the Nb/La proxy is used to evaluate subduction zone vs MORB-type basalts. I also see in the supplement that while low La/ Yb is generally characteristic for MORB-type tholeiites and most B-1 basalts follow this trend, calc-alkaline basalts (and many B-2 basalts) can also have low “tholeiite-like” La/Yb. So I wonder how robust using these geochemical proxies for tectono-magmatic setting is...

In this study, the criterion of $(\text{Nb}/\text{La})_{\text{PM}}$ ratio was extracted from the GEOROC and the Schmidt and Grunder database by using the kernel density statistics (with 95 % credibility) of modern primitive basalts with established tectonic settings, suggesting that $(\text{Nb}/\text{La})_{\text{PM}} \geq 0.75$ is a powerful criteria identifying primitive melts that were generated in the non-subduction settings (e.g., mid-ocean ridge, oceanic island and oceanic plateau), whereas the subduction-related basaltic melts (generated from arc and back-arc basin settings) show $(\text{Nb}/\text{La})_{\text{PM}} < 0.75$ (Supplementary Figure 2). Notably, some initial arc basalts and most of the modern back-arc basin basalts (BABBs) have low “tholeiite-like” $(\text{La}/\text{Yb})_{\text{N}}$ ratios (Schmidt and Grunder, 2009). Although the initial arc basalts and BABBs have similar REE patterns to those of modern tholeiitic MORBs, the most significant difference is that the former has variable degrees of Nb, Ta and Ti depletions (Data from the GEOROC and the Schmidt and Grunder’s database; see the below REE and trace element patterns in this reply). Therefore, we believe that using the basalt $(\text{Nb}/\text{La})_{\text{PM}}$ ratio to reflect tectonic settings is more reliable than the $(\text{La}/\text{Yb})_{\text{N}}$ ratio standard of tholeiitic and calc-alkaline rock series as shown below.

Data of the primitive initial arc basalts and BABBs are from the GEOROC and the Schmidt and Grunder database. The solid green line represents the average values of modern initial arc basalts and BABBs.

Recently, more and more lines of evidence (e.g., the discovery of eclogitic inclusions, the rate of crustal growth and the composition of mafic rocks) indicate that plate subduction or similar mechanism can operate in the Archean (Shirey and

Richardson, 2011; Cawood et al., 2018; Nebel et al., 2018; Hawkesworth et al., 2019; Dien et al., 2020; Sun et al., 2021). Some scholars use the terms of ‘Archean MORB’ and ‘Archean island arc basalt’ based on the basalt La/Yb ratios, and hence it is necessary and urgent to give a more reliable $(\text{Nb/La})_{\text{PM}}$ division to Archean basalts. When dividing the Archean basalts into B-1 and B-2 subgroups, we notice that there are 233 B-2 samples (with a total number of 1507) have $(\text{La/Yb})_{\text{N}}$ ratios < 1.00 , and they exhibit similar REE patterns to B-1 samples (Supplementary Table 1). However, the B-2 samples (with low “tholeiite-like” La/Yb ratios) have significant Nb and Ti depletions with an average $(\text{Nb/La})_{\text{PM}}$ ratio of 0.61, distinct to the negligible Nb and Ti anomalies in B-1 samples with an average $(\text{Nb/La})_{\text{PM}}$ ratio of 0.91. Therefore, given the current situation, we believe the $(\text{Nb/La})_{\text{PM}}$ division to Archean basalt and tectonic setting is reliable. See lines 88-119.

Cited references:

Schmidt, M. E. & Grunder, A. L. The evolution of North Sister: A volcano shaped by extension and ice in the central Oregon Cascade Arc. *Geol. Soc. Am. Bull.* 121, 643-662 (2009).

Cawood, P. A. et al. Geological archive of the onset of plate tectonics. *Phil. Trans. R. Soc. A* 376, 20170405 (2018).

Nebel O, Capitanio F, Moyen J-F, Weinberg R, Clos F, Nebel-Jacobsen YJ, Cawood PA. When crust comes of age: on the chemical evolution of Archaean, felsic continental crust by crustal drip tectonics. *Phil. Trans. R. Soc. A* 376, 20180103 (2018).

Hawkesworth, C., Cawood, P. A. & Dhuime, B. Rates of generation and growth of the continental crust. *Geosci. Front.* 10, 165-173 (2019).

Shirey, S. B. & Richardson, S. H. Start of the Wilson Cycle at 3 Ga shown by diamonds from subcontinental mantle. *Science* 333, 434-436 (2011).

Dien, H. G. E., Doucet, L. S., Murphy, J. B. & Zheng, X. L. Geochemical evidence for a widespread mantle re-enrichment 3.2 billion years ago: implications for global-scale plate tectonics. *Sci. Rep.* 10, 9461 (2020).

Sun, G. et al. Thermal state and evolving geodynamic regimes of the Meso- to Neoproterozoic North China Craton. Nat. Commun. 12, 3888 (2021).

9. Additionally, there may be issues of basalts preserving “pristine” upper mantle redox, because degassing and S might play a role in modifying the basalt fO_2 . There have been instances of finding correlations between redox-sensitive elements and S in OIBs (work of Kelley and Cottrell), so the problem of degassing and its effects on redox is problematic for basalts (which are generated through both lower T_s and increments of melting vs. komatiites). Of course, using komatiites in redox proxies may also have drawbacks, but the authors should offer a somewhat more balanced evaluation of the justification/methodology here.

Aulbach et al. (2019) has recently discussed the influence of degassing or interaction with polyvalent gas species (e.g., S) on the fO_2 of basalt magmas, as shown below: *“Depending on their pressure of emplacement and the nature of the volatile species, degassing of basalts can increase or decrease the $Fe^{3+}/\Sigma Fe$ and hence redox state inferred for the magma. Recent work finds no evidence that degassing or interaction with polyvalent gas species, such as S, has affected $Fe^{3+}/\Sigma Fe$ in modern MORBs, nor that fO_2 is externally buffered, and we suggest that this also applies to magma emplacement in palaeo-ridges. For degassing to be important, differences in process between the Archean and today would be required. However, even if degassing had affected $Fe^{3+}/\Sigma Fe$, such changes in valence state do not change elemental redox proxies, such as V/Sc, the ratio of which in the undifferentiated magma is set at source.”* Therefore, we believe that there may be no direct evidence that these changes would influence the V-Ti redox proxy and V/Ti ratio in basaltic magmas. In this study, we only use basalts with $MgO \geq 8.5$ wt.% to avert possible effects from the late overprint of fractional crystallization and contamination from continental material. We have added a balanced evaluation in the updated manuscript, together with the discussion on some drawbacks of using komatiites. See lines 46-51 and 132-138.

Cited references:

Aulbach, S. et al. Evidence for a dominantly reducing Archaean ambient mantle from two redox proxies, and low oxygen fugacity of deeply subducted oceanic crust. *Sci. Rep.* 9, 20190 (2019).

10. 70: perhaps more sensitive “than” instead of “to”

Revised. See line 73.

11. 114: reference for the software?

Revised. See line 140.

12. 124: What is lower “level” conditions? I would argue the “lower level” P-T for B-2 are not significantly lower at all – in fact the P-T ranges overlap completely, although B-1 does extend to higher T and P.

Revised. See lines 144-149.

13. 136 – 155-ish: This section is contradictory and does not accurately represent the results reported on Fig2. I would agree with the authors that in general, the basalts (if they accurately represent their mantle sources) demonstrate the upper mantle has a heterogeneous fO_2 in the Archean, which appears on average to be reduced by about 1 log unit compared to modern MORB. Basalts from some locations (e.g. Dharwar) appear to show a “progressive” oxidation, but then other locations show opposite trends: that is progressive reduction with younging (Superior). In summary, the B-1 basalts, which are presumed to represent the upper mantle, do not clearly show a temporal evaluation of mantle fO_2 and I believe that a statistical analysis of the data points in Fig2a would not show any clear trends. The secular oxidation as represented

by the B-2 samples is much more apparent – but this is probably because they are derived from heterogeneous sources that may have experienced progressively more input of recycled oxidized sources.

Thanks for your comment.

Revised.

We have expended the Archean basalt database to include the 172 recently published basalt data, and then modified the parameters of our redox models based on the comment 15 (change the ‘fertile peridotite’ to ‘depleted peridotite’ melting reactions). In the updated text, we process the fO_2 results on the basis of different Archean cratons. Hence, the results of B-1 and B-2 magmas are more reliable and have essentially changed: (1) The oxygen fugacity of both ambient and modified mantle exhibits ~1.0 log unit increase beneath most Archean cratons (three temporal craton groups; Figures 1-3). See more details from the reply to comment 2; (2) The nearly constant starting and final fO_2 values suggest that a homogeneous mantle prior to ~3.8 Ga (Figures 1 and 2). In addition, the mantle redox state may be an effective indicator of craton maturation and cratonization (Figure 2); and, (3) These changes are closely associated with the changes in basalt Th/Nb ratios and Nd isotopes, indicating that the gradual oxidation of Archean upper mantle in each craton and craton group is most likely related to recycling of crustal materials into mantle.

14. The uncertainties in the determined fO_2 values are very small – on the order of 0.1 log units. Oxybarometry for mantle-derived rocks generally gives uncertainties at minimum 0.5 log units. In the supplement, there is no discussion regarding the calculation of uncertainties on the fO_2 calculation. Certainly sources of error that arise from the P-T calculations and trace element analyses are propagated into the overall redox composition calculation? A rigorous outline of the uncertainties should be presented, especially since the total variation of the B-1 and B-2 basalts occurs within <2 log units, so some of the “trends” might disappear if the fO_2 uncertainties were to increase.

Thanks for your comment.

Revised.

All of the uncertainties from trace element analyses, mantle compositional heterogeneity, P-T determinations and systematic biases for the Kds of V in mantle minerals and fO_2 are introduced and propagated into the final calculated results of fO_2 . Four aspects of uncertainties are involved in application of V-Ti redox proxy and conversion of V/Ti ratio into mantle fO_2 calculations: (1) Analytical uncertainties of V and Ti were uniformly set to be 5 %, assuming that all samples collected in this study were analyzed by the inductively-coupled plasma mass-spectrometry (ICP-MS) technique (Nicklas et al., 2019; Wang et al., 2019); (2) Uncertainties of mantle heterogeneity were taken as 7 % for V and 12 % for Ti, respectively (Wang et al., 2019); (3) Uncertainties of P-T determinations are based on an absolute value of 0.20 GPa and 3 %, respectively (Lee et al., 2009), assuming that there were no additional uncertainties in the water content assessment; (4) Estimates of uncertainties and systematic biases for the functions between Kds of V in mantle minerals and fO_2 were incorporated into further propagations (Wang et al., 2019). In this study, the uncertainties (1) and (2) were firstly propagated into the calculation of the compositions of V and V/Ti in the melts and their Kds in bulk rock compositions, which were then propagated to the Kds in mantle minerals and fO_2 on the basis of a full account of uncertainties (3) and (4). On this basis, the average propagated 1SD uncertainties of the mantle fO_2 revealed by B-1 and -2 samples are 0.40 and 0.32 log units, respectively. See lines 156-173.

Compared to the uncertainties obtained by the behavior of V in komatiites (about 0.10-0.20 log units in 1SD; Nicklas et al., 2019), our results have considered more error sources of the natural mantle melting and are significantly higher than that obtained from other calculated methods of fO_2 , indicating that the determination of uncertainty and error propagation used in this study is reliable. In addition, the mantle oxidation tendencies with ~1.0 log unit increase beneath most Archean cratons and temporal craton groups still exist according to the propagated errors of 0.40 and 0.32

log units for ambient and modified mantle, respectively (Figures 1-3).

Cited references:

Nicklas, R. W. et al. Secular mantle oxidation across the Archean-Proterozoic boundary: Evidence from V partitioning in komatiites and picrites. *Geochim. Cosmochim. Acta* 250, 49-75 (2019).

Wang, J. et al. Oxidation State of Arc Mantle Revealed by Partitioning of V, Sc, and Ti Between Mantle Minerals and Basaltic Melts. *J. Geophys. Res. Solid Earth* 124, 4617-4638 (2019).

Lee, C.-T. A., Luffi, P., Plank, T., Dalton, H. & Leeman, W. P. Constraints on the depths and temperatures of basaltic magma generation on Earth and other terrestrial planets using new thermobarometers for mafic magmas. *Earth Planet. Sci. Lett.* 279, 20-33 (2009).

15. Also pertaining to the calculation of the fO_2 – the melt modelling equations are conducted on garnet and spinel lherzolites – from “primitive mantle”. For the Eoarchean basalts this seems appropriate, but some of the younger basalts that formed after production of more voluminous continental crust (towards the end of the Archean) might be better modelled using a “depleted mantle” type peridotite.

Thanks for your good suggestion.

Revised.

Considering that the earliest continental crust on the Earth may form during ~4.4-4.2 Ga, therefore, the depleted mantle (DM) compositions were uniformly used in our modellings of ~3.8-2.5 Ga basalts. In the updated text, we employ near fractional melting to perform ~1 GPa (Model A) and ~3 GPa (Model B) dry peridotite melting for B-1 samples, and ~1-2 GPa (Model C) and ~3 GPa (Model D) melting for B-2 samples under hydrous condition, respectively. The initial V and Ti contents and mineral compositions referred to the DM compositions (Salters and Stracke, 2004). In model A, the melting reactions of anhydrous depleted peridotite at 1.0 GPa were used

in this model (Wasylenki et al., 2003). In models C and D, the melting reactions performed by a depleted peridotite metasomatized by a MORB-derived hydrous silicate melt at low pressure (~1-2 GPa) and high pressure (~3 GPa), respectively (Lara and Dasgupta, 2020). However, we have not found previous studies in melting of depleted peridotite at high pressure (~3 GPa). According to our calculations, the fO_2 results obtained by melting models of fertile and depleted peridotite at low pressure (~1-2 GPa) are nearly consistent, for example, sample 43832 has ΔFMQ -0.83 from melting of the fertile peridotite and ΔFMQ -0.89 from melting of the depleted peridotite, respectively. Therefore, the melting reactions of dry garnet peridotite at 3 GPa are used in model B (Walter, 1998). See lines 299-332.

Cited references:

Salters, V. J. M., & Stracke, A. Composition of the depleted mantle. *Geochem. Geophys. Geosyst.* 5, Q05B07 (2004).

Workman, R. K. & Hart, S. R. Major and trace element composition of the depleted MORB mantle (DMM). *Earth Planet. Sci. Lett.* 231, 53-72 (2005).

Lara, M. & Dasgupta, R. Partial melting of a depleted peridotite metasomatized by a MORB-derived hydrous silicate melt-Implications for subduction zone magmatism. *Geochim. Cosmochim. Acta* 290, 137-161, (2020).

Wasylenki, L. E., Baker, M. B., Kent, A. J. R. & Stolper, E. M. Near-solidus Melting of the Shallow Upper Mantle: Partial Melting Experiments on Depleted Peridotite. *J. Petrol.* 1163-1191 (2003).

Walter, M. J. Melting of Garnet Peridotite and the Origin of Komatiite and Depleted Lithosphere. *J. Petrol.* 39, 29-60 (1998).

16. “Increase in crustal recycling” section: This section comes as a sharp contrast to the rest of the paper. While more “space” could have been used to fully evaluate the variability of upper mantle fO_2 with geochemical (or even isotopic) proxies to tease out the type and proportion of crustal contamination or mantle reservoir. Instead the paper turns to a statistical breakdown of the GEOROC Archean mafic rock database

and some geochemical discrimination diagrams. Rather large conclusions are made from this statistical breakdown: “non subduction geodynamic regime play a dominant role” from 3.8 to 3.2 Ga based on % of basalt and komatiite. Firstly, the proportion of komatiite seems very high – it is well known from greenstone belts on cratons that the proportion of komatiite is quite small. How were the proportions (e.g. 48% for komatiite between 3.2 – 3.8 Ga) calculated? Surely not from a simple % from the GEOROC database entries? I did not see any information in the Methods or Supplement. This needs to be fully discussed because the komatiites seem very high. Secondly, such statements about the operation of subduction in the Archean need to be better supported, using perhaps complimentary evidence from other studies that have further investigated Archean rocks for “signs” of subduction tectonics. Take your pick here: Shirey, Dhuime, Kamber, etc. Then we come to Figure 3d-f which models basalt source geochemistry based on some trace element ratios. This type of figure and argument have been made by many papers in the past, and it is not clear how or why this fits with the theme of secular oxidation proposed by the title and introduction of this paper. Obviously the more geochemically enriched B-2 basalts tend towards more “crustal” inputs...it would have been more interesting to do such modelling in terms of redox, considering what we know of Archean recycled oceanic crust as represented by eclogite xenolith studies by Aulbach et al (op. cit) and Smart et al. 2017, 2021a,b.

Thanks a lot.

Revised.

In the last version, we did overestimate of the proportion of komatiites in the worldwide greenstone belts without full consideration of bias sampling. In fact, the komatiites are only < 10 % of the exposed Archean volcanic rocks. As you suggested, the original discussion of rock type statistics (including the estimated proportion of komatiites) and geochemical modelling on the sources of B-1 and B-2 samples may not fit with this paper. We agree with your suggestion, and the related contents have been deleted in the updated text.

The updated text has now focused on the redox state of Archean ambient and

modified mantle (see lines 174-191), the fO_2 variations and implications beneath Archean cratons and temporal craton groups during Eo- to Neoproterozoic (see lines 195-234), the sources and mechanisms of mantle oxidation (see lines 236-270), and the potential geodynamic regime (see lines 271-276). Then, we have considered the evidence of plate subduction or similar mechanisms from other studies as you pointed out. See lines 271-276.

Your final suggestion of geochemical modelling in terms of redox with incompatible elements and eclogitic xenolith studies is suggestive, however, in current situation, there may be some difficulties in model selection, parameter determination and data filtration. We fully agree that these ideas are better to describe in another longer format paper, exploring the mantle geochemical evolution of the early Earth. We will do this research in the near future, and wish a further cooperation and communication. Thanks again for your comments, leading to great improvements in the quality of this manuscript.

Response to comments of Reviewer #3 (Prof. Wenjiao Xiao)

This contribution reports the redox evolution of Archean mantle mainly based on the mantle-derived volcanic rocks (Archean basalts) on Earth. Their updated redox proxies suggest that the Archean mantle was undergoing a secular oxidation process, owing to increased crustal recycling via plate subduction. This study adds to our understanding of Earth's early evolution by providing insight into the elemental evolution of the early Precambrian mantle.

I still have reservations regarding the derivation from the raw data to the conclusions, hence I suggest a minor revision of this work before publishing, taking into account the issues listed below.

1. This discovery is based primarily on data from Earth's Archean basalts. My first concern is how basalt ages were determined. Are the ages accurate, given that dating basic volcanic rocks is typically difficult?

Thank you for this comment. Our basalt data is mainly collected from the

GEOROC database, and the dating methods listed in this database are mainly composed of the whole-rock Rb-Sr, Sm-Nd, Re-Os and Pb-Pb isotopic dating, and the magmatic zircon U-Pb isotopic dating. In this study, we apply the following principles in determining the formation ages of the Archean basalts: (1) The magmatic zircon U-Pb isotopic, whole-rock Sm-Nd and Re-Os isotopic systematics are commonly used to mafic rocks, and believed to be accurate and not easily susceptible to late thermal events. We also carefully check the analysis process and data quality of standard materials, and only select those samples that meet the analysis criteria, for example, the reliable ages obtained by Wan et al. (2011), Guo et al., (2013, 2017) and Wang et al. (2015) and so on using the magmatic zircon U-Pb isotopic dating method, and Puchtel et al. (2004, 2005, 2007) and so on using the Sm-Nd and Re-Os isochron method; (2) If isotopic dating is not available, we will then carefully check the description of geological relationship in their original papers, for example, interlayered relationship with felsic rocks or intrusive relationship with granitoids and mafic dykes (Wang et al., 2013; Gao et al., 2019), which can give the lower limit to the formation age. Therefore, the selected samples in this study should have reliable formation ages traced by isotopic dating methods (mainly via whole-rock Sm-Nd, Re-Os and magmatic zircon U-Pb isotopic systematics) or limited by geological relationships. Although the obtained ages may not be completely accurate, there is no better method to obtain more accurate ages of the Archean basalts based on the current research.

Cited references:

- Wan, Y. et al. ~2.7Ga juvenile crust formation in the North China Craton (Taishan-Xintai area, western Shandong Province): Further evidence of an understated event from U–Pb dating and Hf isotopic composition of zircon. *Precambrian Res.* 186, 169-180 (2011).
- Guo, R. et al. Arc-generated metavolcanic rocks in the Anshan–Benxi greenstone belt, North China Craton: Constraints from geochemistry and zircon U–Pb–Hf isotopic systematics. *Precambrian Res.* 303, 228-250 (2017).

Guo, R. et al. Geochemistry, zircon U–Pb geochronology and Lu–Hf isotopes of metavolcanics from eastern Hebei reveal Neoproterozoic subduction tectonics in the North China Craton. *Gondwana Res.* 24, 664–686 (2013).

Wang, W. et al. Neoproterozoic intra-oceanic arc system in the Western Liaoning Province: Implications for Early Precambrian crustal evolution in the Eastern Block of the North China Craton. *Earth-Sci Rev.* 150, 329–364 (2015).

Gao, L. et al. A Ca. 2.8-Ga Plume-Induced Intraoceanic Arc System in the Eastern North China Craton. *Tectonics* 38, 1694–1717 (2019).

Puchtel I. S., Brandon A. D. & Humayun M. Precise Pt-Re-Os Isotope Systematics Of The Mantle From 2.7-Ga Komatiites. *Earth Planet. Sci. Lett.* 224, 157–174 (2004).

Puchtel I. S., Humayun M. & Walker R. J. Os-Pb-Nd Isotope And Highly Siderophile And Lithophile Trace Element Systematics Of Komatiitic Rocks From The Volotsk Suite, Se Baltic Shield. *Precambrian Res.* 158, 119–137 (2007).

Puchtel I. S., Walker R. J., Brandon A. D. & Nisbet E. G. Pt-Re-Os And Sm-Nd Isotope And Hse And Re Systematics Of The 2.7 Ga Belingwe And Abitibi Komatiites. *Geochim. Cosmochim. Acta* 73, 6367–6389 (2009).

Wang, W. et al. Geochemistry of ~2.7Ga basalts from Taishan area: Constraints on the evolution of early Neoproterozoic granite-greenstone belt in western Shandong Province, China. *Precambrian Res.* 224, 94–109 (2013).

2. According to the authors, the oxidation of the Archean upper mantle is due to increased crustal recycling into the mantle, potentially due to plate subduction. While early subduction tectonics on Earth is a hotly debated topic. Is there another mechanism that accounts for the recycling of crustal materials or the oxidation of the mantle? It's possible that more discussion of various options is required.

We have added the related contents in the revised manuscript (lines 271–276). Two major Archean geodynamic models of plate subduction (or hot subduction) and lithospheric drip, which are potential to convey sufficient crustal materials into mantle, have been proposed on the basis of a higher mantle potential temperature than the

present day (van Hunen and van den Berg et al., 2008; Cawood et al., 2018; Capitanio et al., 2019; Gao et al., 2019). The lithospheric drip model has been recently established under a series of thermal numerical modellings. The uppermost lithosphere in this model is assumed to be nearly motionless and its lower layer is dragged by convective mantle to form some dip drips, which appear like a similar pattern of plate subduction, providing an optional and potential mechanism of crustal recycling.

Therefore, we consider that the crustal recycling, perhaps via plate subduction or drip tectonics, could be the most powerful mechanism leading to gradual upper mantle oxidization in the Archean.

Cited references:

Gao, L. et al. A Ca. 2.8-Ga Plume-Induced Intraoceanic Arc System in the Eastern North China Craton. *Tectonics* 38, 1694-1717 (2019).

Capitanio F.A., Nebel O., Cawood P.A., Weinberg R.F. & Chowdhury P. Reconciling thermal regimes and tectonics of the early Earth. *Geology* 47, 923-927 (2019).

Cawood P.A., Hawkesworth C.J., Pisarevsky S.A., Dhuime B., Capitanio F.A. & Nebel O. Geological archive of the onset of plate tectonics. *Philosophical Transactions of the Royal Society A: Mathematical Physical and Engineering Sciences* 376, 20170405 (2018).

van Hunen, J. & van den Berg, A. P. Plate tectonics on the early Earth: Limitations imposed by strength and buoyancy of subducted lithosphere. *Lithos* 103, 217-235 (2008).

Bédard J.H. Stagnant lids and mantle overturns: Implications for Archean tectonics, magmagenesis, crustal growth, mantle evolution, and the start of plate tectonics. *Geoscience Frontiers* 9, 19-49 (2018).

Moyen J.F. & Laurent O. Archean tectonic systems: A view from igneous rocks. *Lithos* 302-303, 99-125 (2018).

3. My final question concerns the redox state of the Archean earth's surface. Was it

enough to alter the redox status of the mantle, given that Earth's great oxidation occurred much later?

The geological record of mass-independent sulfur isotope fractionation shows that significant O₂ in the atmosphere occurred during the Great Oxidation Event (GOE) between 2.4 and 2.1 Ga (Lyons et al., 2014). However, redox-sensitive iron and molybdenum isotope data suggest the presence of O₂ in the 3.2-3.0 Ga marine photic zone, which implies that O₂-producing cyanobacteria existed long before the GOE (Planavsky et al., 2014; Satkoski et al., 2015). Similarly, geological data indicate the presence of methanotrophy and oxidative nitrogen cycling in Neoproterozoic oceans and lakes, which suggest that O₂ was oxidizing sulfides and ammonium to make sulfate and nitrate, respectively (Godfrey et al., 2009; Garvin et al., 2009). Indeed, the oxidation time of the Earth's surface remains highly controversial, namely, the GOE might have occurred earlier than previously proposed by the GOE time of the ~2.3 Ga. Notably, based on our previous studies, the *f*O₂ values of Archean crust-derived granitoid magmas are significantly higher (up to FMQ +5.5 at ~2.8-2.7 Ga and FMQ +8.9 at ~2.5 Ga) than those of synchronous mantle (Figures 1 and 2; Hu et al., 2019). Therefore, we consider that the Archean Earth's surface might have the capacity to effect on the mantle redox state via crustal recycling.

Cited references:

Lyons, T. W., Reinhard, C. T. & Planavsky, N. J. The rise of oxygen in Earth's early ocean and atmosphere. *Nature* 506, 307-315 (2014).

Planavsky, N. J. et al. Evidence for oxygenic photosynthesis half a billion years before the Great Oxidation Event. *Nat. Geosci.* 7, 283-286 (2014).

Satkoski, A. M., Beukes, N. J., Li, W. Q., Beard, B. L. & Johnson, C. M. A redox-stratified ocean 3.2 billion years ago. *Earth Planet. Sci. Lett.* 430, 43-53 (2015).

Godfrey, L. V. & Falkowski, P. G. The cycling and redox state of nitrogen in the Archean ocean. *Nat. Geosci.* 2, 725-729 (2009).

Garvin, J., Buick, R., Anbar, A. D., Arnold, G. L. & Kaufman, A. J. Isotopic evidence

for an aerobic nitrogen cycle in the latest Archean. *Science* 323, 1045-1048 (2009).

Hu, Y.L., Liu, S.W., Sun, G.Z. & Gao, L. Petrogenesis of the Neoproterozoic granitoids and crustal oxidation states in the Western Shandong Province, North China Craton. *Precambrian Research* 334, 105446 (2019).

REVIEWERS' COMMENTS

Reviewer #3 (Remarks to the Author):

The revised manuscript assembled a global whole-rock geochemical database of Archean basalts, applied an updated V-Ti redox proxy, and ultimately succeeded in offering new insights into resolving Archean upper mantle oxidation depending on the individual data processing and calculation. After carefully reading the revised manuscript, the authors addressed the concerns I raised previously, including demonstrating the accuracy of raw geochronologic data from basic volcanic rocks, presenting alternative mechanisms for crustal recycling, and demonstrating the effect of the Earth's surface's redox state. This document contains no evident logical errors or inconsistencies. As a result, I believe this manuscript is acceptable for publication.

Wenjiao Xiao

Reviewer #4 (Remarks to the Author):

Dear Editors. I read through the revised manuscript and the original reviews plus response by the authors. Overall, the criticism raised has been addressed, for most part. Of course, working with such ancient rocks and database entries will always retain an element of speculation, but I think the authors argue well within the boundaries of their assumptions.

I have a few minor additional comments:

- 1) The title (also Line 193) reads a bit awkward with that "beneath cratons" part. Why not just leaving that part out? In the end, the authors work with basaltic rocks formed between 3.8 Ga and 2.5 Ga, which are only preserved on cratons, but they could have formed anywhere (if considered to represent the ambient upper mantle). So I think that such a simplification of the title is fair and would highlight the global relevance of this study better.
- 2) Lines 32, 84 etc: it should say 'changes in Nd isotope ratios (or compositions). Just saying "changes in Nd isotopes" reads a little sloppy.
- 3) Line 204-206, and elsewhere: The North Atlantic and Kaapvaal cratons are viewed as having a "similar evolutionary history between 3.8 and 3 Ga", based on the apparently similar redox evolution of basaltic magmas (and their sources) preserved on (or beneath) these cratons. This statement is somewhat 'sweeping' because from actual studies of the mantle roots of these two cratons we know that they fall on opposite ends of the geochemical evolution in terms of peridotite Mg-Si relations etc (e.g., Pearson & Wittig, 2014, for a review). That is, the NAC mantle is highly melt-depleted and dominated by dunitic lithologies, whereas the Kaapvaal mantle has received strong SiO₂-additions during the Archean (by subduction zone processes?), as expressed by an overabundance of orthopyroxene in most peridotite xenoliths. So the Archean evolution of these two cratons is far from similar and this requires some more careful wording.
- 4) Lines 237-238: nitrogen isotopes should be added to this listing, with credits to the study by Smart et al. (2016, Nature Geoscience). The same reference is also highly pertinent to, and supportive of, the lithospheric plate subduction argument in terms of mantle redox evolution, as mentioned in Line 272.
- 5) Line 274: "bring recognized"? Not sure if this is a proper English expression – please check with Dr. Cawood.
- 6) The manuscript cites a lot of studies, but given that during the revision process constraints from komatiites have been largely abandoned, the authors may want to save on references referring to komatiites (when the redox context is missing).

Good luck!

Sebastian Tappe

UiT The Arctic University of Norway

Response to referees letters

Response to comments of Reviewer #4

Dear Editors. I read through the revised manuscript and the original reviews plus response by the authors. Overall, the criticism raised has been addressed, for most part. Of course, working with such ancient rocks and database entries will always retain an element of speculation, but I think the authors argue well within the boundaries of their assumptions.

I have a few minor additional comments:

- 1) The title (also Line 193) reads a bit awkward with that “beneath cratons” part. Why not just leaving that part out? In the end, the authors work with basaltic rocks formed between 3.8 Ga and 2.5 Ga, which are only preserved on cratons, but they could have formed anywhere (if considered to represent the ambient upper mantle). So I think that such a simplification of the title is fair and would highlight the global relevance of this study better.

Revised. We have changed original title ‘Oxidation of Archean upper mantle beneath cratons caused by crustal recycling’ into ‘Oxidation of Archean upper mantle caused by crustal recycling’. See the updated title.

- 2) Lines 32, 84 etc: it should say ‘changes in Nd isotope ratios (or compositions)’. Just saying “changes in Nd isotopes” reads a little sloppy.

Revised. Original usage of ‘changes in Nd isotopes’ has been changed into ‘changes in Nd isotope ratios’. See lines 31 and 82.

3) Line 204-206, and elsewhere: The North Atlantic and Kaapvaal cratons are viewed as having a “similar evolutionary history between 3.8 and 3 Ga”, based on the apparently similar redox evolution of basaltic magmas (and their sources) preserved on (or beneath) these cratons. This statement is somewhat ‘sweeping’ because from actual studies of the mantle roots of these two cratons we know that they fall on opposite ends of the geochemical evolution in terms of peridotite Mg-Si relations etc (e.g., Pearson & Wittig, 2014, for a review). That is, the NAC mantle is highly melt-depleted and dominated by dunitic lithologies, whereas the Kaapvaal mantle has received strong SiO₂-additions during the Archean (by subduction zone processes?), as expressed by an overabundance of orthopyroxene in most peridotite xenoliths. So the Archean evolution of these two cratons is far from similar and this requires some more careful wording.

Thank you for your suggestion.

In recent studies, more and more hornblende-bearing peridotites with harzburgitic normative composition (the hornblende was most likely generated by replacement of orthopyroxene and olivine during the secondary alteration) have been found in the North Atlantic craton (Löcht et al., 2020). Geochemical and petrogenetic studies indicate that these mantle peridotites can provide direct evidence for the siliceous melt additions and the preservation and operation of Eoarchean subduction processes (Löcht et al., 2020). The majority of the peridotite xenoliths in the Kaapvaal craton have consistent Mg-Si relations to those of peridotite in the North Atlantic craton (See the left figure below. Compositions of these mantle peridotites are collected from the GEOROC database).

The basalt Mg-Si relations in these two cratons are also identical (See the right figure below. Basalt compositions are collected from the GEOROC database). In this study, the proportions of B-1 and B-2 samples in the North Atlantic Craton are 15 % and 85 %, respectively, nearly consistent with those in the Kaapvaal Craton of 7 % and 93 %, respectively. Similarly, many studies based on field geology, and igneous and metamorphic petrology, indicate that the upper mantle beneath both

of these cratons experienced strong crustal recycling, possibly via plate subduction or other similar mechanism (Polat and Hofmann, 2003; Nutman et al., 2009; Turner et al., 2014; Nutman et al., 2020, 2021; Agangi, 2020; Nakamura et al., 2020; Guotana et al., 2021). Therefore, we consider that the mantle beneath the North Atlantic and Kaapvaal cratons through that time record a similar history.

The Mg-Si relations of mantle peridotites (left) and basalts (right) in the North Atlantic and Kaapvaal cratons (Data from the GEOROC database).

Cited references:

van de Locht, J., Hoffmann, J. E., Rosing, M. T., Sprung, P., Münker, C., 2020. Preservation of Eoarchean mantle processes in ~3.8 Ga peridotite enclaves in the Itsaq Gneiss Complex, southern West Greenland. *Geochimica et Cosmochimica Acta* 280, 1-25.

Guotana, J.M., Morishita, T., Nishio, I., Tamura, A., Mizukami, T., Tani, K., Harigane, Y., Szilas, K., Pearson, D.G., 2021. Deserpentinization and high-pressure (eclogite-facies) metamorphic features in the Eoarchean ultramafic body from Isua, Greenland, *Geoscience Frontiers* 2021.101298.

Nakamura, H., Sano, A., Kagami, S., Yokoyama, T., Ishikawa, A., Komiya, T., Iwamori, H., 2020. Compositional heterogeneity of Archean mantle estimated from Sr and Nd isotopic systematics of basaltic rocks from North Pole, Australia, and the Isua supracrustal belt, Greenland. *Precambrian Research* 347, 105803.

Nutman, A.P., Friend, C.R.L., Paxton, S., 2009. Detrital zircon sedimentary provenance ages for the Eoarchean Isua supracrustal belt southern West

Greenland: Juxtaposition of a ca. 3700 Ma juvenile arc assemblage against an older complex with 3920-3800 Ma components. *Precambrian Research* 172, 212-233.

Nutman, A.P., Bennett, V.C., Friend, C.R.L., Keewook Yi, 2020. Eoarchean contrasting ultra-high-pressure to low-pressure metamorphisms (< 250 to >1000 °C/GPa) explained by tectonic plate convergence in deep time. *Precambrian Research* 344, 105770.

Nutman, A.P., Scicchitano, M., Friend, C.R.L., Bennett, V.C., Chivas, A.R., 2021. Isua (Greenland) ~3700 Ma Ultra High Pressure mantle meta-peridotite olivine Mg# and $\delta^{18}\text{O}$ signatures show connection between the early mantle and hydrosphere: Geodynamic implications. *Precambrian Research*, 361, 106249.

Polat, A., Hofmann, A.W., 2003. Alteration and geochemical patterns in the 3.7-3.8 Ga Isua greenstone belt, West Greenland. *Precambrian Research* 126, 197-218.

Turner, S., Rushmer, T., Reagan, M., Moyen, J.-F., 2014. Heading down early on? Start of subduction on Earth. *Geology* 42, 139-142.

Agangi, A., 2020. The Mesoarchean Dominion Group and the onset of intracontinental volcanism on the Kaapvaal craton-Geological, geochemical and temporal constraints. *Gondwana Research* 84, 131-150.

- 4) Lines 237-238: nitrogen isotopes should be added to this listing, with credits to the study by Smart et al. (2016, *Nature Geoscience*). The same reference is also highly pertinent to, and supportive of, the lithospheric plate subduction argument in terms of mantle redox evolution, as mentioned in Line 272.

Revised. We have added this reference (Smart et al., 2016) in its appropriate locations. See lines 231-232 and 267-268.

- 5) Line 274: “bring recognized”? Not sure if this is a proper English expression – please check with Dr. Cawood.

Revised. This sentence has been revised into ‘Although our data alone cannot differentiate between these mechanisms, it does establish that Archean crustal recycling was widespread and ongoing within each craton over hundreds of millions of years’. See lines 267-268

- 6) The manuscript cites a lot of studies, but given that during the revision process constraints from komatiites have been largely abandoned, the authors may want to save on references referring to komatiites (when the redox context is missing).

Revised. We have further deleted some references referring to komatiites without redox context, for example, Kerrich and Xie (2002) and Sossi et al. (2016). At present, the number of references is 57. See the updated list of references.